# Damage-induced reactive oxygen species regulate *vimentin* and dynamic collagen-based projections to mediate wound repair

Danny LeBert[1,2†], Jayne M Squirrell[3†], Chrissy Freisinger[2], Julie Rindy[2], Netta Golenberg[2], Grace Frecentese[3], Angela Gibson[4], Kevin W Eliceiri[3], Anna Huttenlocher[2,5*]

[1]Department of Biology, Shenandoah University, Winchester, United States; [2]Department of Medical Microbiology and Immunology, University of Wisconsin-Madison, Madison, United States; [3]Laboratory for Optical and Computational Instrumentation, University of Wisconsin-Madison, Madison, United States; [4]Department of Surgery, University of Wisconsin-Madison, Madison, United States; [5]Department of Pediatrics, University of Wisconsin-Madison, Madison, United States

**Abstract** Tissue injury leads to early wound-associated reactive oxygen species (ROS) production that mediate tissue regeneration. To identify mechanisms that function downstream of redox signals that modulate regeneration, a *vimentin* reporter of mesenchymal cells was generated by driving GFP from the *vimentin* promoter in zebrafish. Early redox signaling mediated *vimentin* reporter activity at the wound margin. Moreover, both ROS and vimentin were necessary for collagen production and reorganization into projections at the leading edge of the wound. Second harmonic generation time-lapse imaging revealed that the collagen projections were associated with dynamic epithelial extensions at the wound edge during wound repair. Perturbing collagen organization by burn wound disrupted epithelial projections and subsequent wound healing. Taken together our findings suggest that ROS and vimentin integrate early wound signals to orchestrate the formation of collagen-based projections that guide regenerative growth during efficient wound repair.

DOI: https://doi.org/10.7554/eLife.30703.001

*For correspondence: huttenlocher@wisc.edu

†These authors contributed equally to this work

Competing interests: The authors declare that no competing interests exist.

## Introduction

Wound repair requires the integration of cellular signaling networks to efficiently restore tissue homeostasis. Response to tissue injury can lead to either scar formation or complete tissue regeneration, which occurs in certain teleosts and amphibians (*Wehner and Weidinger, 2015*). Understanding repair in regenerative animals can help inform these processes in humans and other species where scar formation is the primary response to injury. Recent studies highlight the importance of early signaling events after damage, including the role of reactive oxygen species (ROS), in tissue regeneration in zebrafish and tadpoles (*Gauron et al., 2013*; *Love et al., 2013*; *Yoo et al., 2012*). In tadpoles, injury induced ROS activate growth factor signaling pathways through Wnt and fibroblast growth factor (fgf) that mediate regeneration (*Love et al., 2013*). Inhibiting early ROS signaling at damaged epithelium for just 1 hr pre-and post-injury in larval zebrafish impairs subsequent fin regeneration 3 days post-injury (*Yoo et al., 2012*). Although a role for Src family kinase signaling has been implicated as a pathway downstream of this early ROS signaling (*Yoo et al., 2012*), the redox-dependent pathways that mediate full regeneration in zebrafish larvae remain unclear.

Movement of epithelial cells is thought to play key roles in ensuring proper wound healing (*LeBert and Huttenlocher, 2014*; *Sonnemann and Bement, 2011*). During this migration, epithelial cells undergo a process known as epithelial to mesenchymal transition (EMT), a progression conserved during normal development as well as the malignant transformation of epithelial cells (*De Craene and Berx, 2013*). The intermediate filament protein vimentin is expressed during EMT and provides a marker of EMT during cell transformation (*De Craene and Berx, 2013*). Although well characterized as a marker of EMT, how vimentin regulates wound healing is still unclear. A recent report demonstrated that vimentin regulates wound healing by affecting fibroblast proliferation and the differentiation of keratinocytes via transforming growth factor beta (TGFβ) and Slug signaling (*Cheng et al., 2016*). Additionally, vimentin has been implicated in regulating cell movements during wound repair, as seen with vimentin's role in the collective movements of lens epithelium after damage (*Menko et al., 2014*). In glial cells, vimentin polarizes toward the wound edge, in collaboration with the microtubule network (*Leduc and Etienne-Manneville, 2017*). In recent studies, vimentin was shown to orient actin filaments and traction stress during single cell migration (*Costigliola et al., 2017*). These recent studies suggest an important role for vimentin in regulating the movement of cells during wound repair.

In addition to cell movement, a critical stage of normal repair is the deposition of a collagen-rich extracellular matrix (ECM) to provide the framework for regenerative growth. The transient ECM that forms during wound healing is subsequently degraded during the remodeling phase of wound healing (*McCarty and Percival, 2013*). Collagen accumulates in response to tissue injury and it appears vimentin is required for this accumulation, likely indirectly as suggested by the deficiency of wound-associated fibroblasts in vimentin null mice (*Cheng et al., 2016*; *Eckes et al., 2000*). In addition to the production of collagen, a key step in the repair process is the remodeling of wound-induced collagen structures, probably through the action of proteases such as matrix metalloproteinases (MMP). In zebrafish larvae we recently reported a central role for MMP9 in collagen reorganization and regeneration after tail transection (*LeBert et al., 2015*).

Here we sought to determine the pathways that function downstream of wound-associated ROS that mediate wound healing and subsequent regrowth. We generated a reporter of mesenchymal cells by driving GFP from the *vimentin* promoter and found that early redox signaling at the wound was required for *vimentin* expression at the wound margin. We also found that inhibition of ROS, NFκB or depletion of vimentin impaired both *collagen* expression at the wound and the dynamic reorganization of collagen into projections. Live imaging demonstrated that these collagen projections guide epithelial regrowth during regeneration of the fin. Treatments that disrupt collagen organization, such as inhibition of cross-linking or destruction of collagen fibers by thermal injury, diminished the formation of epithelial projections and subsequent wound healing was impaired. These results provide a pathway linking early ROS signaling to the physical and mechanical processes of healing and regrowth. This healing process is mediated by vimentin and its regulation of collagen fiber/epithelial cell projections that promote the forward progression of the wound plane and lead to the regeneration of the caudal fin.

## Results

### Generation of a *vimentin* reporter line in zebrafish

Previous studies demonstrated that loss of vimentin impaired wound healing in mice (*Eckes et al., 2000*), suggesting that vimentin plays a role in tissue repair. To determine how *vimentin* is regulated in response to tissue injury in zebrafish larvae, we generated a reporter of *vimentin* expression by driving EGFP from the *vimentin* promoter, herein referred to as the *Tg(−2vim:egfp)* line (*Figure 1*). Reporter activity was detected as early as the two somite stage (*Figure 1A*) and later observed in early differentiating neurons in the head and in glial cells within the developing spinal cord (*Figure 1B*). From 2–3 days post fertilization (dpf) the EGFP expression pattern was maintained in the cranial ganglion cells, spinal cord neurons and opercle (*Figure 1C,D,E,F*). This expression pattern is consistent with the *vimentin* expression pattern previously shown by in situ hybridization (*Cerdà et al., 1998*). We also noted EGFP expression in developing fin mesenchymal cells at 3 dpf (*Figure 1C,G*). It is known that vimentin expression increases with the induction of epithelial to mesenchymal transition (EMT) (*Franke et al., 1982*; *Thiery, 2002*; *Vuoriluoto et al., 2011*). To examine

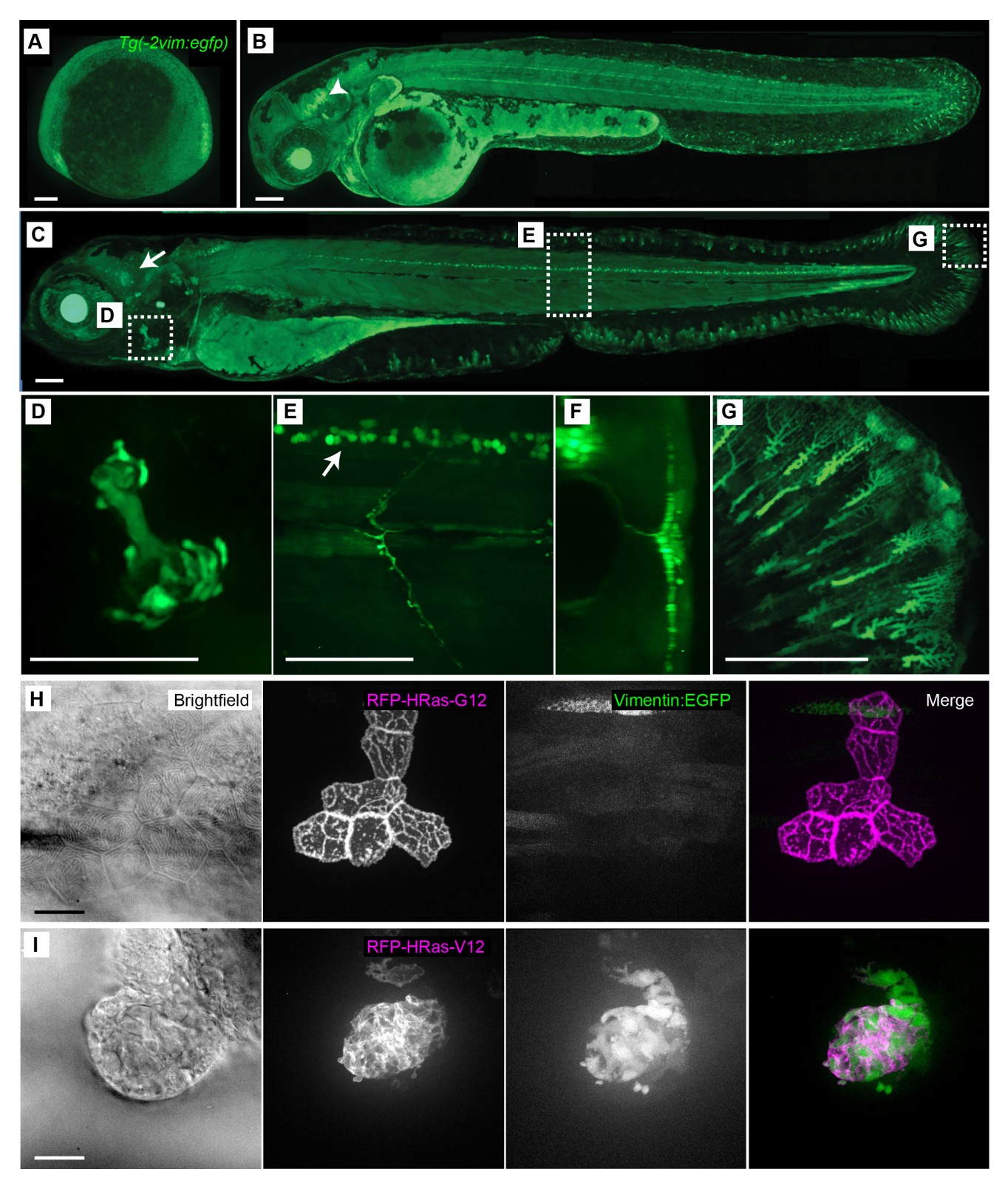

**Figure 1.** Characterization of the *Tg(−2vim:egfp)* zebrafish line. The *Tg(−2vim:egfp)* larvae express EGFP in cranial ganglion cells, spinal cord neurons, the opercle and fin fibroblasts. Images depict lateral views of *Tg(−2vim:egfp)* larvae with anterior to the left of (A) two somite stage, (B) 2 dpf, arrow indicating cranial ganglion cells, and (C) 4 dpf *Tg(−2vim:egfp)* larvae, arrow indicating EGFP positive cranial ganglion cells while boxes show regions used for subsequent zoomed images. (D) EGFP expression in the opercle. (E) and (F) spinal cord neuron expression, enface and cross section,

*Figure 1 continued on next page*

*Figure 1 continued*

respectively. (**G**) Mesenchymal cells of the caudal fin. (**H**) Mosaic RNA expression of HRas[G12] in the *Tg(−2vim:egfp)* line did not induce *vimentin* promoter activation. (**I**) Mosaic RNA expression of oncogenic HRas[V12] in the *Tg(−2vim:egfp)* line resulted in a dramatic increase in *vimentin* promoter activation, observed in two replicates. Scale bars represent 100 μm in A-G and 20 μm in H-I.

DOI: https://doi.org/10.7554/eLife.30703.002

the expression of this reporter during EMT we transiently expressed either wild type (HRas[G12]) or oncogenic RAS (HRas[V12]), tagged with RFP, from the *krt4* promoter in the *Tg(−2vim:egfp)* line (*Figure 1H,I*), thereby inducing EMT (*Freisinger and Huttenlocher, 2014*). Overexpression of oncogenic HRas[V12], but not wild type HRas[G12], induced *vimentin* promoter activity and expression of EGFP that co-localized in the same cells expressing HRas[V12]. Taken together, these data show that the *Tg(−2vim:egfp)* line recapitulated previously published in situ *vimentin* expression data and vimentin was expressed in cells undergoing EMT, thereby providing a powerful tool to report *vimentin* promoter activity in larval zebrafish.

## *vimentin* reporter activity is induced by tissue injury downstream of redox signaling

Since vimentin plays a key role in wound healing (*Cheng et al., 2016*; *Eckes et al., 2000*; *Menko et al., 2014*), we sought to determine if *vimentin* expression is increased at the wound. To address this question, we performed whole mount in situ hybridization (WMISH) to probe *vimentin* expression in unwounded and wounded larvae and found that tail transection induced localized *vimentin* expression at the wound edge by 24 hr post wound (hpw), an outcome not observed with the sense probe (*Figure 2A–C*, *Figure 2—figure supplement 1A*). To further characterize the population of cells that express *vimentin* at the wound edge (vim+), we utilized long duration time-lapse laser scanning confocal microscopy of living *Tg(−2vim:egfp)* reporter larvae. Interestingly, in the unwounded fin, the reporter line labeled a population of elongated mesenchymal cells that extend toward the tip of the fin (*Figure 1G*). Since the majority of these vim+ cells in the fin were removed during the tail transection no clear signal was detected along the wound edge at 2 hpw (*Figure 2D*). In accordance with the in situ results, vim+ cells were detected by 20 hpw at the wound edge and showed enhanced expression at 26 hpw (*Figure 2E–G*). To determine if this population of vim+ cells at the wound edge resulted from the migration of distal vim+ cells, we performed long duration time-lapse laser scanning confocal microscopy in a controlled environment using the zWEDGI platform that is compatible with long term live imaging (*Huemer et al., 2017*). Surprisingly, the vim+ cells did not migrate to the wound from surrounding fin tissue but rather expression developed in what appeared to be a non-motile population of cells at the wound edge (*Figure 2—video 1*). It should be noted that these vim+ cells did not appear to arise from epithelial cells based on live imaging (*Figure 2—figure supplement 1B and C*, *Figure 2—video 2*). However, these data do not preclude the possibility that the vim+ cells are the result of an EMT event beyond the time resolution of our studies. Additionally, the vim+ cells did not appear to colocalize with a macrophage marker (*Figure 2—figure supplement 1D*) and their presence was not influenced by the depletion of leukocytes using the *pu.1* morpholino (MO) (*Figure 2—figure supplement 1E*), suggesting that immune cells were not involved directly or indirectly in the vimentin expression. It is possible that a specialized population of cells, including a type of progenitor cell, migrates to the wound edge from the surrounding tissue and subsequently activates the *vimentin* promoter, although our studies cannot confirm this possibility. Regardless, our findings demonstrate that wounding induces a population of vim+ cells along the wound edge.

Wound-induced release of ROS has been shown to be a key regulator of wound healing and regeneration (*Leisegang et al., 2016*; *Serras, 2016*; *Yoo et al., 2012*). Our earlier work demonstrated that early ROS signaling immediately before and after damage, mediated fin regeneration three days later (*Yoo et al., 2012*). Therefore, we sought to determine whether early wound-induced ROS mediate *vimentin* expression at the wound. Using a pharmacological inhibitor of NAD(P)H oxidase enzyme activity, Diphenyleneiodonium (DPI), we found that inhibition of ROS for 1 hr before and 1 hr after wounding was sufficient to impair the appearance of vim+ cells at the wound edge 1 dpw in the *Tg(−2vim:egfp)* line (*Figure 2H,I*). Reduced reporter activity after tail transection with

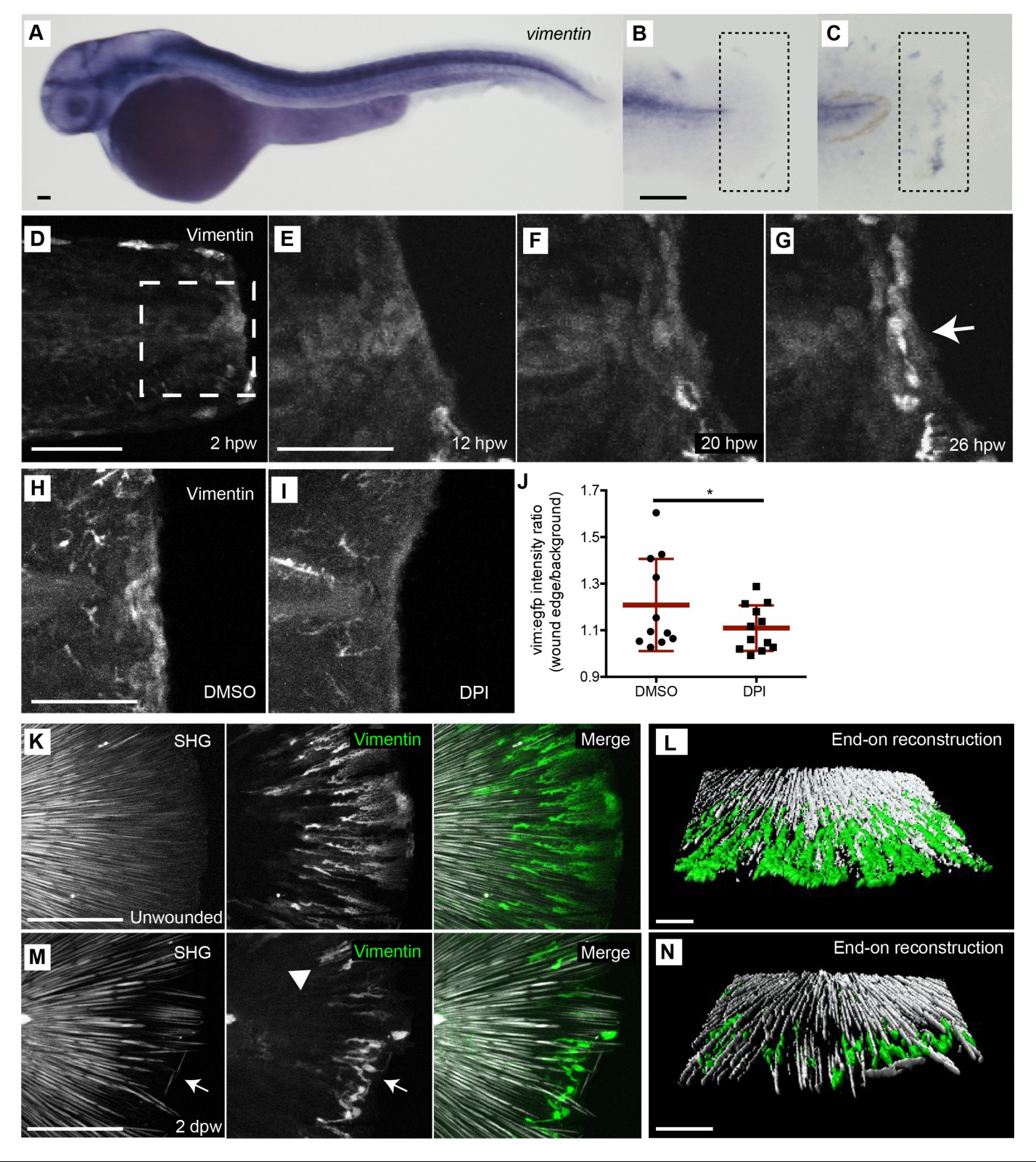

**Figure 2.** A population of vimentin-positive cells with a rounded morphology developed at the wound edge and associate with areas of collagen misalignment. (**A**) Whole mount in situ hybridization for *vimentin* RNA in 4 dpf unwounded larvae. (**B**) Unwounded caudal fin and (**C**) 24 hpw caudal fin indicates increased *vimentin* expression following amputation, observed in two experiments. (**D-G**) Confocal microscopy of a live larval caudal fin showed that following amputation in the *Tg(−2vim:egfp)* line, in the wound region (box), a population of cells activated the *vimentin* promoter (arrow) *Figure 2 continued on next page*

*Figure 2 continued*

by 26 hpw. (H,I) The activation of the *vimentin* promoter was regulated by ROS, as treatment with DPI reduced EGFP signal at the wound edge at 24 hpw as compared to control DMSO. (J) Ratio of the intensity of the vim:egfp expression at the central wound edge over background intensity of the fin in DPI and DMSO treated larvae at 24 hpw. (p=0.0282; n = 12 total fins over three replicates, with 2 to 6 fins per treatment per replicate; DMSO 95% CI = 1.18 to 1.33, DPI 95% CI = 1.07 to 1.21). (K) Spindle-shaped vim+ cells associate tightly with SHG fibers, as indicated by enface 3D reconstruction of SHG images of unwounded caudal fins in the *Tg(−2vim:egfp)* line. (L) End on view of a 3D surface rendered reconstruction of the association of vim+ cells and SHG fibers in unwounded caudal fins. (M) Spindle shaped vim+ cells associate with intact SHG fibers following amputation (Arrowhead), but a rounded vim+ cell population is observed in regions of collagen misalignment (Arrow) in the *Tg(−2vim:egfp)*. (N) End on surface rendered reconstruction of the association of vim+ cells and SHG fibers in amputated caudal fins at 2 dpw, assessed in two replicates including 24 live larvae. Scale bar represents 100 μm in A-C, 50 μm in D-I, K and M and 30 μm in L and N. *p<0.05, error bars are standard deviation.
DOI: https://doi.org/10.7554/eLife.30703.003

The following video, source data, and figure supplement are available for figure 2:

**Source data 1.** Command sequence used in R for LSMeans analysis.
DOI: https://doi.org/10.7554/eLife.30703.005
**Source data 2.** Vimentin GFP intensity at the wound in DPI treated larvae.
DOI: https://doi.org/10.7554/eLife.30703.006
**Figure supplement 1.** Vimentin expression at the wound edge did not co-localize with epithelial or macrophage markers and was not influenced by leukocyte depletion.
DOI: https://doi.org/10.7554/eLife.30703.004
**Figure 2—video 1.** *vimentin* promoter is activated following amputation in the *Tg(−2vim:egfp)* line.
DOI: https://doi.org/10.7554/eLife.30703.007
**Figure 2—video 2.** Vim(+) cells do not appear to co-localize with Krt4tdTom(+) cell.
DOI: https://doi.org/10.7554/eLife.30703.008
**Figure 2—video 3.** Vim(+) cells aligned along SHG fibers in an unwounded caudal fin.
DOI: https://doi.org/10.7554/eLife.30703.009
**Figure 2—video 4.** Vim(+) cells exhibit a rounded morphology in the presence of disrupted SHG fibers after tail transection.
DOI: https://doi.org/10.7554/eLife.30703.010

early ROS inhibition was quantified by assessing the relative fluorescence intensity above background at the wound edge (*Figure 2J*, *Figure 2—source data 1*, *Figure 2—source data 2*). Similar results were observed with morpholino depletion of Duox1 (*Yoo et al., 2012*), which is necessary for the generation of wound-induced ROS (*Figure 2—figure supplement 1F,G*). Previous studies using a transgenic nuclear factor kappa-light-chain-enhancer of activated B cells (NFκB) activation reporter *Tg(nfκb:egfp)* line showed that early ROS signaling regulates NFκB activity following caudal fin amputation (*de Oliveira et al., 2014*). This led us to investigate if ROS was regulating *vimentin* activation through NFκB. Using a pharmacological inhibitor of NFκB activation, Withaferin A (WA), which has previously been shown to inhibit caudal fin wound healing (*LeBert et al., 2015*), we found that early inhibition of NFκB was sufficient to block EGFP expression at 1 dpw in the *Tg(−2vim:egfp)* line (*Figure 2—figure supplement 1*). It should be noted that WA can directly inhibit the vimentin protein, and thus the influence of WA downstream of *vimentin* promoter activation must also be considered. However, the findings suggest that early ROS signaling, likely at least in part through NFκB activity, regulates *vimentin* promoter activation at 1 dpw following caudal fin amputation.

The radiating, elongated morphology of the vim+ cells in the unwounded fin was reminiscent of the pattern previously reported for collagen fibers in the developing zebrafish fin using second harmonic generation (SHG) imaging (*LeBert et al., 2015*). To address a possible association between the vim+ cells and collagen fibers, we performed SHG imaging of the caudal fin in the *Tg(−2vim: egfp)* line. We observed a tight association between the vim+ cells and the SHG fibers (*Figure 2K,L*) in unwounded caudal fins (*Figure 2—video 3*). In wounded caudal fins we observed vim+ cells with a more rounded morphology in regions of the fin characterized by mis-aligned SHG collagen fibers and the elongated cells in regions of the fin with intact collagen fibers (*Figure 2M,N*, *Figure 2— video 4*), indicating an association between vim+ cells and organization of the SHG detected collagen fibers.

## Vimentin is required for optimal caudal fin wound healing in zebrafish

The increased *vimentin* expression observed at the wound edge suggests that an increase in vimentin may be necessary for optimal wound healing in the larval zebrafish. To determine whether

vimentin is required for larval caudal fin regeneration, we used morpholino depletion of *vimentin* expression (*Figure 3—figure supplement 1A*). We found that vimentin-deficient larvae had impaired regenerate area and regenerate length at 3 dpw (*Figure 3A,B*, *Figure 3—figure supplement 1B*, *Figure 3—source data 1*), indicating that vimentin is required for proper wound healing. This defect was wound-specific, as unwounded morphants did not display a defect in fin length or area compared to age matched controls during normal development (*Figure 3—figure supplement 1F–H*). These findings are consistent with the wound defect reported with VIM-/- mice (*Cheng et al., 2016*; *Eckes et al., 2000*; *Menko et al., 2014*). It should also be noted that vimentin-deficient larvae displayed no defect in the appearance of vim+ cells at the wound edge following amputation (*Figure 3—figure supplement 2A*), indicating that vimentin is not required for the activation of the *vimentin* promoter along the wound edge. To determine if the regenerative defect was due to effects on cell survival, we assessed apoptosis by TUNEL staining. No defect was observed in the vimentin-deficient larvae at 1 hpw (*Figure 3—figure supplement 2B*), providing evidence that the wound healing defect is likely not due to changes in cell survival. To confirm that the regeneration defect was not due to off-target effects of the MO, we rescued *vimentin* expression using co-injection of *vimentin* MO1 and zebrafish *vimentin* RNA and also used a second MO (MO2) (*Figure 3—figure supplement 1D,E*). Co-injection was sufficient to rescue the regeneration defect observed in the morphants, suggesting that vimentin is required for fin regeneration in zebrafish (*Figure 3—figure supplement 3A–C*). As further confirmation that vimentin is required for optimal wound healing, we used CRISPR-Cas9 mutagenesis to create mosaic F0 *vimentin* mutants. The CRISPR was validated by TOPO cloning and sequencing of individual F0 larvae as well as by sequencing germline DNA isolated from F0 adults (*Figure 3C–E*). The mosaic F0 mutants also displayed reduced regenerate length and area at 3 dpw (*Figure 3F,G*, *Figure 3—figure supplement 3D*, *Figure 3—source data 2*) but had no developmental defect in fin length (*Figure 3—figure supplement 3E–G*). In accordance with previous studies in other species (*Cheng et al., 2016*; *Eckes et al., 2000*; *Menko et al., 2014*), these data show that vimentin is required for efficient wound healing.

## Vimentin and early ROS signaling mediate the formation of SHG-fiber containing projections during wound healing

To determine how vimentin regulates wound repair, we performed long duration laser scanning confocal microscopy of wound healing dynamics. During regeneration of the caudal fin following amputation, the wound edge formed an uneven border that contained growing epithelial cell extensions (*Figure 4—video 1*). The formation of these epithelial extensions, herein referred to as projections, was not present during normal development of the caudal fin, which occurred with the extension of a smooth fin edge (*Figure 4—figure supplement 1A*). The first visible projections after tail transection could be identified at 18 hpw and became prominent by 24 hpw (*Figure 4A*). The projections continued to extend and new projections formed throughout wound repair (*Figure 4A*). By contrast, vimentin-deficient larvae possessed a smooth wound edge and relative absence of the epithelial projections for more than 24 hpw (*Figure 4B*, *Figure 4—video 2*). To quantify these effects, we assessed the percentage of larvae that form visible projections. Following transection, more than 75 percent of control larvae had visible projections by 24 hpw while only about 40 percent of vimentin-deficient larvae displayed projections (*Figure 4C*, *Figure 4—source data 1*). Since *vimentin* expression is dependent on early ROS signaling at the wound, we also characterized the effect of early ROS inhibition on the formation of epithelial projections. Early pharmacological inhibition of ROS resulted in a smooth wound edge (*Figure 4—figure supplement 1B*) and impaired the formation of projections at 24 hpw (*Figure 4D*, *Figure 4—source data 2*). Similar effects were observed with inhibition of NFκB using the small molecule WA (*Figure 4—figure supplement 1C*).

To further characterize the epithelial projections and the role of vimentin in this process, we performed bright field imaging of the projections at high magnification. We observed thin finger-like projections at the wound edge that extended around fibrillar proteins that protruded into the tips of the projections (*Figure 4E,F*). This observation of fibrillar structures was supported by LC-PolScope microscopy (*Oldenbourg, 2005*) suggesting that the fibrillar protein was a collagen structure (*Keikhosravi et al., 2017*). LC-PolScope, although similar to traditional polarized light microscopy, uses near circular polarized light that permits high sensitivity detection of fibers in all directions. These fibers appeared to induce a physical stress on the epithelial projections since the epithelial cells, whose membranes are also birefringent and can be detected with LC-PolScope microscopy,

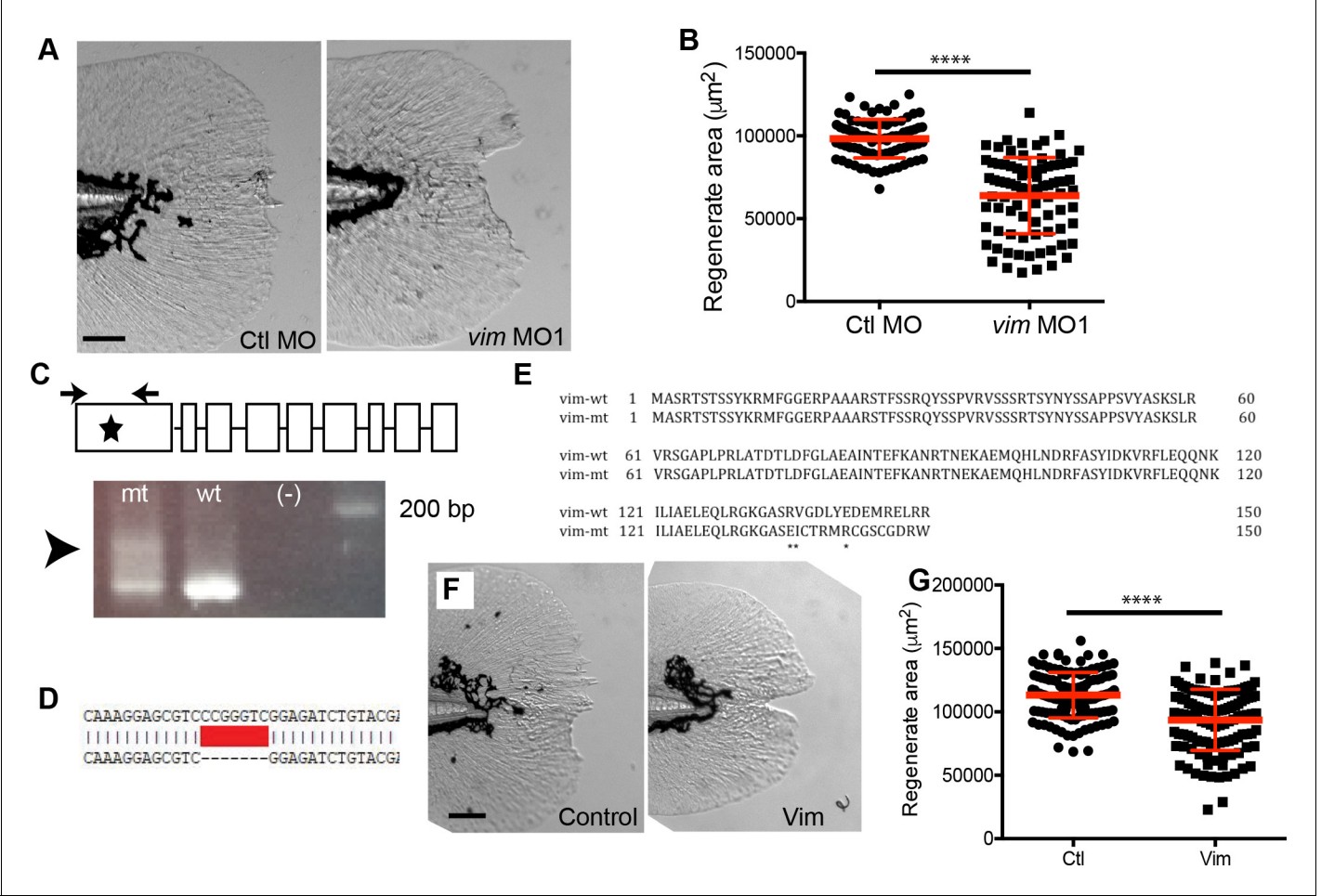

**Figure 3.** Vimentin expression was required for proper wound healing. (A) Morpholino knockdown (MO1) of *vimentin* caused a significant wound healing defect in (B) regenerate area at 3 dpw (p<0.0001; n = 85 total larvae per treatment over four replicates, with 17 to 30 larvae per treatment per replicate; Ctl MO 95% CI = 94604.23 to 101336.98; *vim* MO 95% CI = 60550.63 to 67369.68). (C) Schematic of single-site CRISPR-Cas9 targeting of *vimentin* (top). Arrowhead indicates the presence of aberrant banding in the mosaic F0 mutant (bottom). Sequencing of the *vim* guide RNA target sites was also performed (see Materials and methods) on single embryos to further validate. (D) One example of sequencing results shows a 7 bp deletion in Exon 1, an early stop codon. (E) Amino acid alignment of *vimentin* from germline genomic DNA from adult F0 mosaic adults. (F) Mosaic F0 mutant larvae (Vim) displayed a significant defect in (G) regenerate area at 3 dpw (p<0.0001; ctl n = 130 larvae total, Vim n = 119 larvae total over 3 replicates with 36 to 45 larvae per treatment per replicate; ctrl 95% CI = 110643.20 to 115689.47, Vim 95% CI = 90275.48 to 95555.25). Scale bars in A and F represent 100 µm. Statistical significance: ****p<0.0001, error bars are standard deviation.
DOI: https://doi.org/10.7554/eLife.30703.011

The following source data and figure supplements are available for figure 3:

**Source data 1.** Regenerate area measurements with vimentin knockdown.
DOI: https://doi.org/10.7554/eLife.30703.015

**Source data 2.** Regenerate area measurements with vimentin transient Crispr.
DOI: https://doi.org/10.7554/eLife.30703.016

**Figure supplement 1.** Vimentin depletion reduced *vimentin* expression but did not affect fin development.
DOI: https://doi.org/10.7554/eLife.30703.012

**Figure supplement 2.** Vimentin depletion did not affect *vimentin* promoter activation or apoptosis.
DOI: https://doi.org/10.7554/eLife.30703.013

**Figure supplement 3.** Vimentin morpholino regeneration defect was rescued by *vimentin* RNA and recapitulated in transient mutants.
DOI: https://doi.org/10.7554/eLife.30703.014

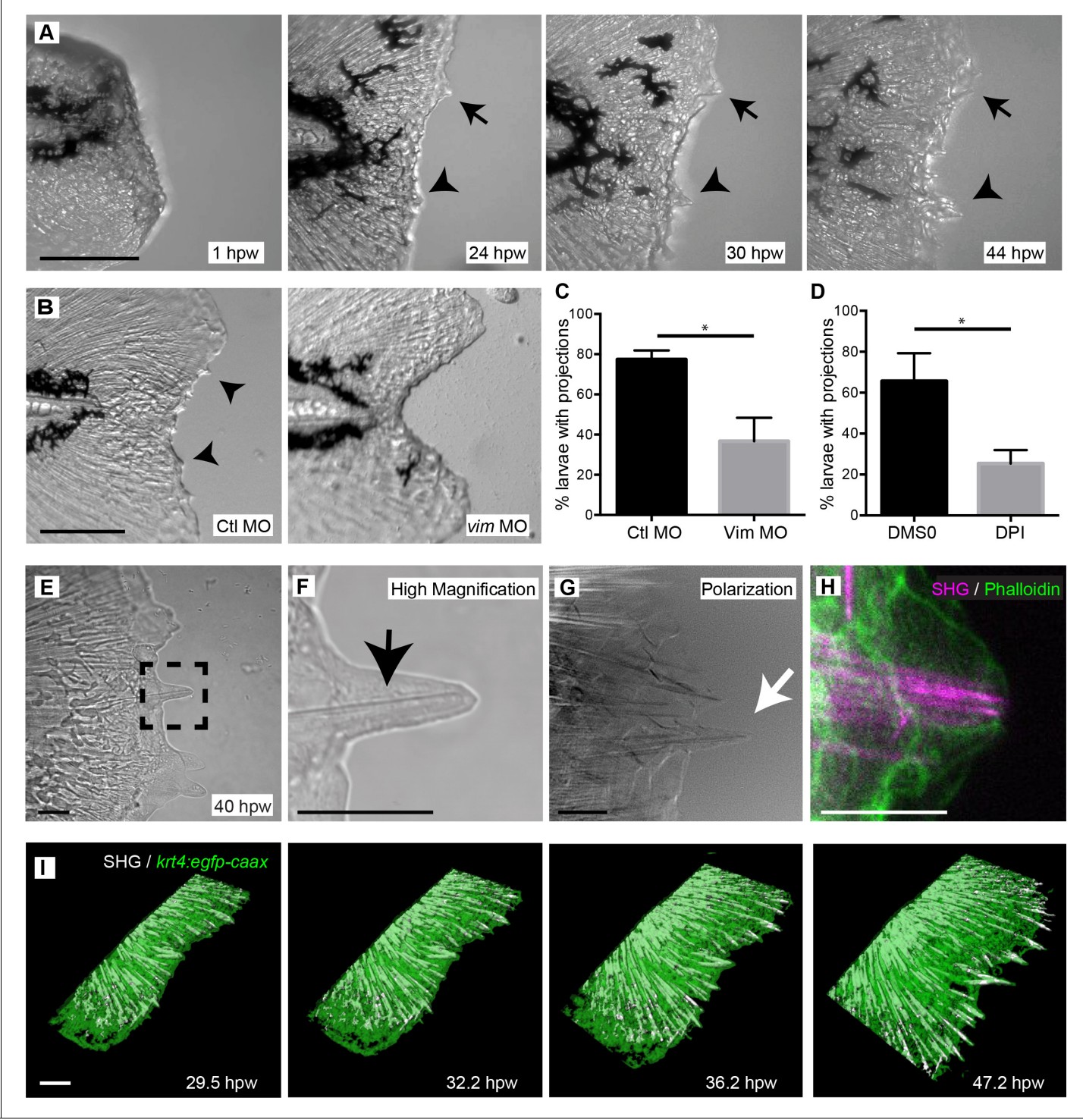

**Figure 4.** Larval zebrafish regenerated the caudal fin following amputation and extended projections. (**A**) Time-lapse microscopy of caudal fin regeneration revealed the formation of epithelial projections by 24 hpw. Projections extending from the amputated plane are indicated with an arrow and arrowhead. (**B, C**) Morpholino knockdown of *vimentin* expression led to a significant reduction in the proportion of larvae forming projections at 24 hpw compared to control larvae (p=0.0348; n = 3 replicate percentages: for ctl MO 76 larvae total and for Vim MO 78 larvae total were scored over the three replicates, with 18 to 34 larvae per treatment per replicate; Ctl MO 95% CI = 66.13% to 88.54%; *vim* MO 95% CI = 7.56% to 65.78%). (**D**) Inhibition of early ROS signaling with DPI treatment also significantly reduced the proportion of larvae forming wound healing projections at 24 hpw (p=0.0129, n = 3 replicate percentages: for DMSO 72 larvae total and for DPI 66 larvae total were scored over the three replicates, with 11 to 26 larvae per treatment per replicate; DMSO 95% CI = 31.76% to 99.58%, DPI 95% CI = 8.79% to 41.87%). (**E, F**) Bright field microscopy of projections revealed the presence of

*Figure 4 continued on next page*

*Figure 4 continued*

fibrillar structures (box, black arrow). (G) Polarization microscopy was used to identify the fibers as organized structures, likely collagen, and by showing the cell membranes, confirmed that the fibers extended to the tip of the projections (white arrow). (H) The association of the fibers and overlying cells was further verified by combining SHG with phalloidin staining of actin. (I) 3D reconstruction surface rendering from time-lapse microscopy, collecting SHG and fluorescence signals in the *Tg(krt4:egfp-caax)* line, showed extending SHG fibers pushing the healing plane forward. Scale bars represent 100 μm in A and B,10 μm in E-H and 30 μm in I. *p<0.05, error bars are standard deviation.

DOI: https://doi.org/10.7554/eLife.30703.017

The following video, source data, and figure supplement are available for figure 4:

**Source data 1.** Proportion of larvae exibiting visible projections in vimentin-deficient larvae.
DOI: https://doi.org/10.7554/eLife.30703.019
**Source data 2.** Proportion of larvae exibiting visible projections in DPI treated larvae.
DOI: https://doi.org/10.7554/eLife.30703.020
**Figure supplement 1.** Pharmacological inhibition of early ROS and NFκB signaling disrupted the formation of SHG and epithelial projections.
DOI: https://doi.org/10.7554/eLife.30703.018
**Figure 4—video 1.** Wound healing progresses following caudal fin amputation with the formation of epithelial projections by 24 hpw
DOI: https://doi.org/10.7554/eLife.30703.021
**Figure 4—video 2.** Wound healing projections are impaired in vimentin-deficient larvae
DOI: https://doi.org/10.7554/eLife.30703.022
**Figure 4—video 3.** Time-lapse microscopy showing that SHG fibers form projections that push out cells thus promoting forward extension of the wound edge during tail regrowth.
DOI: https://doi.org/10.7554/eLife.30703.023
**Figure 4—video 4.** 3D reconstruction of time-lapse microscopy showing SHG fibers pushing epithelial cells forward promoting forward extension of the wound edge during tail regrowth.
DOI: https://doi.org/10.7554/eLife.30703.024

appear elongated, with the cell membranes stretched outward around the fiber (*Figure 4G*). To further characterize these projections, we used SHG microscopy. Live SHG imaging after tail transection confirmed the presence of fibers in the projections and indicated a close association between the fibers and the projecting epithelial cells. Notably, the SHG fibers appeared to push the epithelial cells outward during extension of the wound edge (*Figure 4—video 3*). These findings suggest that the SHG fibers in the caudal fin, which we have previously shown are important for proper wound healing (*LeBert et al., 2015*), play a guidance role in the formation of epithelial projections during wound repair. This was more clearly shown by dual imaging of SHG fibers and cell borders, using both phalloidin staining on fixed samples (*Figure 4H*) or by performing live imaging of the *Tg(krt4: egfp-caax)* line (*Figure 4I*; *Figure 4—video 4*). Inhibition of projection formation in the vimentin-deficient, DPI- and WA treated larvae resulted in a lack of SHG fiber-containing projections as indicated by polarization imaging (*Figure 4—figure supplement 1D, F and H*) and SHG imaging in combination with phalloidin staining (*Figure 4—figure supplement 1E, G and I*).

## Wound projections associate with collagen fibers during wound repair

Since the depletion of vimentin reduced the appearance of epithelial projections (*Figure 4B,C*), we investigated the effect of vimentin depletion on SHG fibers. We found that fiber projections at the wound edge were reduced in vimentin-deficient larvae (*Figure 5A,D*, *Figure 5—videos 1–4*). To quantify this, the length of the wound edge (contour) in SHG images of the region immediately adjacent to the end of the notochord was measured in control and vimentin-depleted larvae (*Figure 5B and E*). The vimentin-deficient larvae had a reduced contour length at the wound edges as compared to control larvae (*Figure 5G*, *Figure 5—source data 1*), supporting the hypothesis that the fiber projections, as well as the epithelial projections, as assessed by bright field imaging, require vimentin.

Because our previous work indicates that SHG fiber organization plays an important role in regrowth of the transected tail fin (*LeBert et al., 2015*), we sought to further examine how vimentin depletion impacts SHG fiber organization. Visual observation of the control MO and *vimentin* MO wounded caudal fins suggested that fibers in the vimentin-depleted fins did not project perpendicularly outward and were prone to be more tangential to the wound edge. To quantify this observation we utilized CurveAlign analysis (*Bredfeldt et al., 2014*; *Liu et al., 2017*; *Schneider et al., 2013*) of

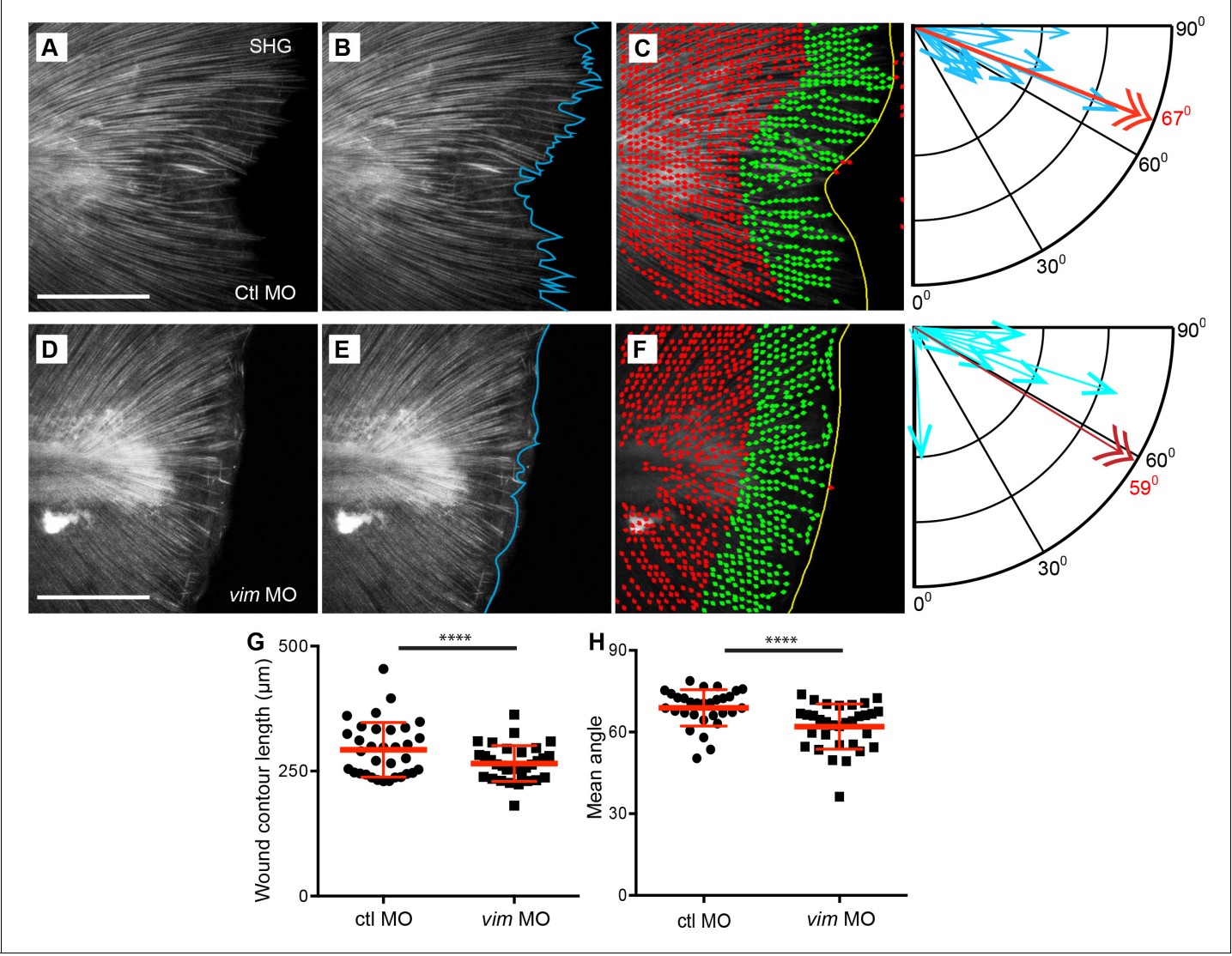

**Figure 5.** Extension of collagen fibers required vimentin. SHG microscopy showed the formation of SHG projections in (A) control, and that the projections were impaired in the (D) *vimentin* morphants. To quantify this observation, Z projections of the region posterior to the notochord, at 24 hpw, were used for analysis. (B, E) The contour of the wound edge was traced and measured, indicated by the blue line. To assess differences in fiber angle relative the wound edge of fibers, the SHG images were quantified using CurveAlign software, which identified fiber segments using curvelet transformation and calculated the angle of each of these segments, within 60 microns of the wound edge, relative to the nearest wound edge (see Methods for further details). The identified segments are illustrated in (C) (control) and (F) (*vimentin* morphant) as dots with small linear extensions, with green being the segments within 60 microns of the wound edge and included in the analysis, as shown in corresponding compass plot, in which the blue arrows indicate the number of segments at a given angle, with 90° being perpendicular and 0° being parallel to the wound edge. The red double arrow and corresponding red numeral indicate the mean angle of the fiber segments for that caudal fin. There was a decrease in (G) wound contour (reduced fiber extensions) in the *vimentin* morphants at 24 hpw compared to controls (p=0.0009; ctl MO n = 33 total larvae, *vim* MO n = 32 total larvae over 4 replicates with 7 to 9 larvae per treatment per replicate; ctrl MO 95% CI = 280.88 to 305.18, *vim* MO 95% CI = 250.32 to 275.1). (H) The fiber angle measurement indicated a significant decrease in orientation, with fibers exhibiting angles more parallel rather than perpendicular, to the wound edge, in the *vimentin* morphants at 24 hpw (p=0.0003; ctl MO n = 30 total larvae, *vim* MO n = 32 total larvae over 4 replicates with 6 to 9 larvae per treatment per replicate; ctl MO 95% CI = 66.12 to 71.36, *vim* MO 95% CI = 59.21 to 64.31). Scale bars represent 100 μm in A and D. ****p<0..0001, error bars are standard deviation.

DOI: https://doi.org/10.7554/eLife.30703.025

The following video, source data, and figure supplement are available for figure 5:

**Source data 1.** Wound contour length in vimentin-deficient larvae.
DOI: https://doi.org/10.7554/eLife.30703.027

**Source data 2.** Fiber angle relative to wound edge in vimentin-deficient larvae.

*Figure 5 continued on next page*

Figure 5 continued

DOI: https://doi.org/10.7554/eLife.30703.028
**Figure supplement 1.** Extension of the SHG fibers after wounding was inhibited by DPI and withaferin A (WA) treatment.
DOI: https://doi.org/10.7554/eLife.30703.026
**Figure 5—video 1.** SHG fibers of the caudal fin of an unwounded 3 dpf control larvae were closely aligned, radiating out to a relatively smooth distal edge.
DOI: https://doi.org/10.7554/eLife.30703.029
**Figure 5—video 2.** SHG fibers of the caudal fin of an unwounded 3 dpf vimentin-depleted larva resembled the closely aligned fibers of the control larvae, with a similarly smooth distal edge.
DOI: https://doi.org/10.7554/eLife.30703.030
**Figure 5—video 3.** SHG fibers of the caudal fin of a 1 dpw control larvae were misaligned, with projections forming at the wound edge.
DOI: https://doi.org/10.7554/eLife.30703.031
**Figure 5—video 4.** SHG fibers of the caudal fin of a 1 dpw vimentin-depleted larvae were misaligned, with fibers more parallel to the projections forming at the wound edge.
DOI: https://doi.org/10.7554/eLife.30703.032

the SHG images of the transected tails to determine the angle of the fibers relative to the wound edge (*Figure 5C and F*). This analysis revealed that the population of fibers with vimentin depletion tended to be less perpendicular overall than their control counterparts (*Figure 5H*, *Figure 5—souce data 2*).

To determine if early ROS signaling also affected SHG projections we performed a similar analysis in the presence and absence of DPI. Inhibition of early ROS signaling impaired SHG fiber contour length at 24 hpw (*Figure 5—figure supplement 1A*) as compared to controls. A similar outcome was observed with WA treatment (*Figure 5—figure supplement 1B*). Taken together, these findings suggest that early wound signaling through ROS regulates *vimentin* expression and the subsequent formation of fiber projections during wound healing. These fibers support the movement of epithelial cell projections to promote wound edge progression during the process of wound healing.

## Vimentin and early wound signaling is required for normal wound-induced expression of *col1a1 and col2a1b*

The caudal fin of the larval zebrafish contains structural components identified as actinotrichia (*Dane and Tucker, 1985*) that are composed of both *col1a1* and *col2a1b* (*Durán et al., 2011*). To confirm that the fibers we observed were composed of multiple types of collagen, we performed SHG to visualize type I/III collagen (*Campagnola et al., 2002*; *Mohler et al., 2003*) and antibody labeling of type II collagen (Col II) by immunofluorescence. Unwounded fins displayed a highly structured organization exhibiting both SHG and Col II antibody label (*Figure 6A*). Following amputation, this ordered orientation of fibers was lost in regions adjacent to the amputated plane (*Figure 6B*), however both SHG fibers and Col II labeling still associated in projections and disrupted fibers. Interestingly, we observed strong Col II labeling at the tips of broken SHG fibers (*Figure 6B*). Future studies will be necessary to determine the function of these Col II caps. Regardless, the findings suggest that the SHG fibers associate with Col II fibers during repair.

Since previous studies suggested that vimentin regulates collagen accumulation after wounding in mouse models (*Cheng et al., 2016*), we wanted to determine whether the effects of vimentin on collagen fibers were due, at least in part, to changes in collagen expression. The major fibrillar collagens of the developing caudal fin are *col1a1* and *col2a1b* (*Durán et al., 2011*). Therefore, we performed quantitative real-time polymerase chain reaction (qRT-PCR) to assess expression of *col1a1* and *col2a1b* following caudal fin amputation. To accurately assess the expression at the site of the wound, an approximately 500 µm length of fin was collected from amputated and age-matched unwounded larvae (*Figure 6C*). At 24 hpw, we found an increase in c*ol1a1* and *col2a1b* expression in the fins of wounded larvae (*Figure 6D*, *Figure 6—source data 1*). In vitro studies in mice show that expression of *collagen I* is regulated by vimentin via stabilization of *collagen I* mRNA (*Challa and Stefanovic, 2011*). To determine if vimentin regulates *collagen* expression in our in vivo model as well, we performed qRT-PCR in vimentin-deficient larvae. At 24 hpw, vimentin-deficient larvae displayed a reduction in *col1a1* and *col2a1b* expression at 24 hpw compared to controls (*Figure 6E*, *Figure 6—source data 2*). Furthermore, pharmacological inhibition of early ROS

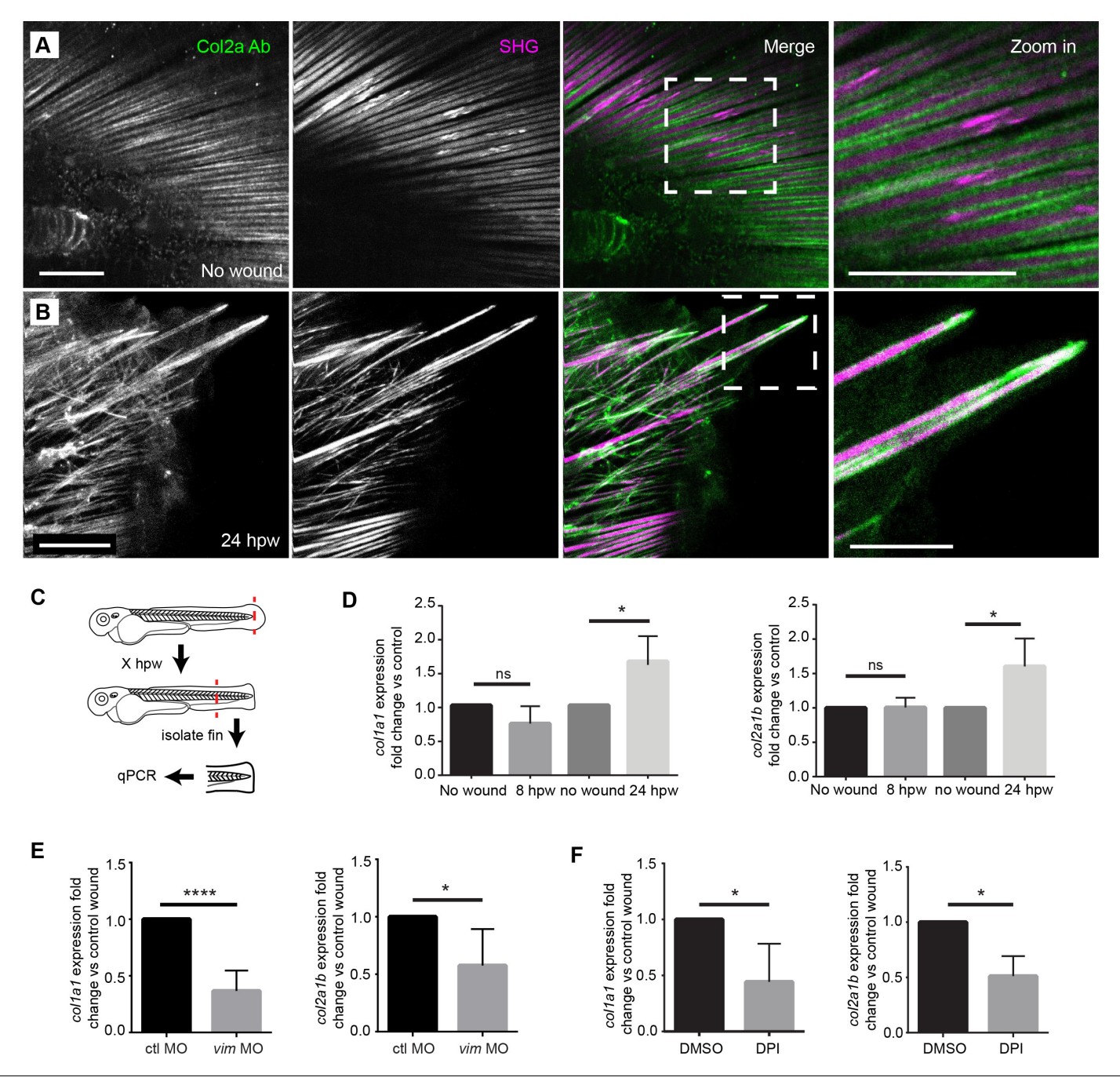

**Figure 6.** Projections contained both SHG fibers and Type II Collagen, and the expression of collagen at wounds was regulated by vimentin and ROS. (A) SHG and type II collagen fibers tightly associate in unwounded caudal fins and (B) within projections at 24 hpw as indicated by SHG imaging and immunofluorescence as observed in two experiments. (C) Schematic of *col1a1* and *col2a1b* expression analysis from the caudal fin. (D) *col1a1* and *col2a1b* expression increased in the wounded caudal fin by 24 hpw as assessed by qRT-PCR (*col1a1* 8 hpw p=0.0779, n = 5 replicates; 24 hpw p=0.0153, n = 6 replicates; col2a1b 8 hpw p=0.9491, n = 3 replicates, 24 hpw p=0.0294, n = 5 replicates). (E) Morpholino knockdown of *vimentin* expression reduced the expression of *col1a1* and *col2a1b* in the wounded caudal fin compared to control (*col1a1* p=0.004, n = 6 replicates; *col2a1b* p=0.0404, n = 5 replicates). (F) Early inhibition of ROS using DPI significantly reduced expression of *col1a1* and *col2a1b* at 24 hpw (col1a1 p=0.0464, n = 4 replicates; col2a1b p=0.0126). *p<0.05. ****p<0.0001; error bars are standard deviation.
DOI: https://doi.org/10.7554/eLife.30703.033

The following source data and figure supplement are available for figure 6:

**Source data 1.** *col1a1* and *col2a1b* expression following wounding.

*Figure 6 continued on next page*

eLIFE Research article

Cell Biology

Figure 6 continued

DOI: https://doi.org/10.7554/eLife.30703.035
**Source data 2.** *col1a1* and *col2a1b* expression following wounding in vimentin-deficient larvae.
DOI: https://doi.org/10.7554/eLife.30703.036
**Source data 3.** *col1a1* and *col2a1b* expression following wounding with DPI treatment.
DOI: https://doi.org/10.7554/eLife.30703.037
**Figure supplement 1.** Larp6-deficient larvae displayed reduced wound induced *collagen* expression.
DOI: https://doi.org/10.7554/eLife.30703.034

signaling was sufficient to cause a significant reduction in the rise of *collagen* expression following amputation at 24 hpw (*Figure 6F*, *Figure 6—source data 3*). A similar result was also observed with WA treatment (*Figure 6—figure supplement 1A*), which may be working either through NFκB inhibition or by affecting vimentin directly.

In vitro analysis of vimentin regulation of *collagen* expression has suggested that vimentin regulates *col1* expression through an interaction with La Ribonucleoprotein Domain Family Member 6 (Larp6) (*Challa and Stefanovic, 2011*). To determine whether this interaction also occurred in our in vivo model, we performed morpholino knockdown of *larp6* (*Figure 6—figure supplement 1B*) followed by qRT-PCR analysis at 24 hpw. We determined that Larp6-deficient larvae have reduced *col1a1* expression but no defect in *col2a1b* expression, consistent with the published in vitro data (*Figure 6—figure supplement 1C*). Furthermore, Larp6-deficient larvae displayed a significant defect in regenerate area at 3 dpw, although no defect in regenerate length (*Figure 6—figure supplement 1D*), suggesting that vimentin is influencing wound healing through multiple pathways.

## Disrupting collagen stability reduced projections and resulted in wound healing defects

Our findings suggest that vimentin affects wound healing, at least in part through affecting collagen fiber extensions and the formation of epithelial projections at the wound edge. To determine if perturbing collagen cross linking affects the formation of projections and wound repair, we used a pharmacological inhibitor of lysyl oxidase, beta aminopropionitrile (BAPN) to inhibit collagen I cross-linking and the formation of collagen I fibrils (*Kagan and Li, 2003*; *Pinnell and Martin, 1968*; *Tang et al., 1983*; *Wilmarth and Froines, 1992*). Constant inhibition of lysyl oxidase throughout the wound healing process caused a significant reduction in regeneration of the larval fin at 2 dpw (*Figure 7—figure supplement 1A–C*). Experiments to 3 dpw were not possible as the inhibitor caused lethality in both wounded and un-wounded larvae between 4 and 5 dpf (data not shown). It is possible that the observed effects on regeneration were in part due to toxicity but it should be noted that no effect on developmental fin length was observed at the concentration that impaired regeneration. We next performed SHG imaging to determine if a defect in collagen fiber projections was also present. Interestingly, BAPN treatment caused an increase in contour length over control (*Figure 7—figure supplement 1D,E*). We believe this to be a consequence of the measurement technique, as the V-shaped fin in the BAPN treated larvae created more contour to measure. However, the BAPN treatment caused a significant decrease in SHG fiber mean angle, consistent with vimentin-deficient larvae (*Figure 7—figure supplement 1F*) and indicates a disruption in fiber organization. Taken together, these findings suggest that collagen cross-linking mediates regeneration of the fin and reorganization of collagen at the wound edge.

## Burn injury impaired collagen projections and wound repair

To further study the connection between the formation of epithelial projections, collagen fiber extensions and healing outcome, we utilized a new zebrafish burn wound assay. Using short bursts of stimulation from a cauterizing wire, we generated a wound to the caudal fin epithelium that also caused significant damage to the integrity of the SHG fibers near the wound edge (*Figure 7A*) such that fibers were absent from the region adjacent to the wound. We performed SHG imaging on phalloidin labeled samples 24 hr post burn (hpb) and again observed a striking absence of fibers in the region near the wound edge in the burned caudal fins (*Figure 7B*). To determine if fins experiencing a burn wound and subsequent loss of collagen fiber integrity along the wound edge impacted healing dynamics, we compared burned caudal fins to age-matched, caudal fin-amputated

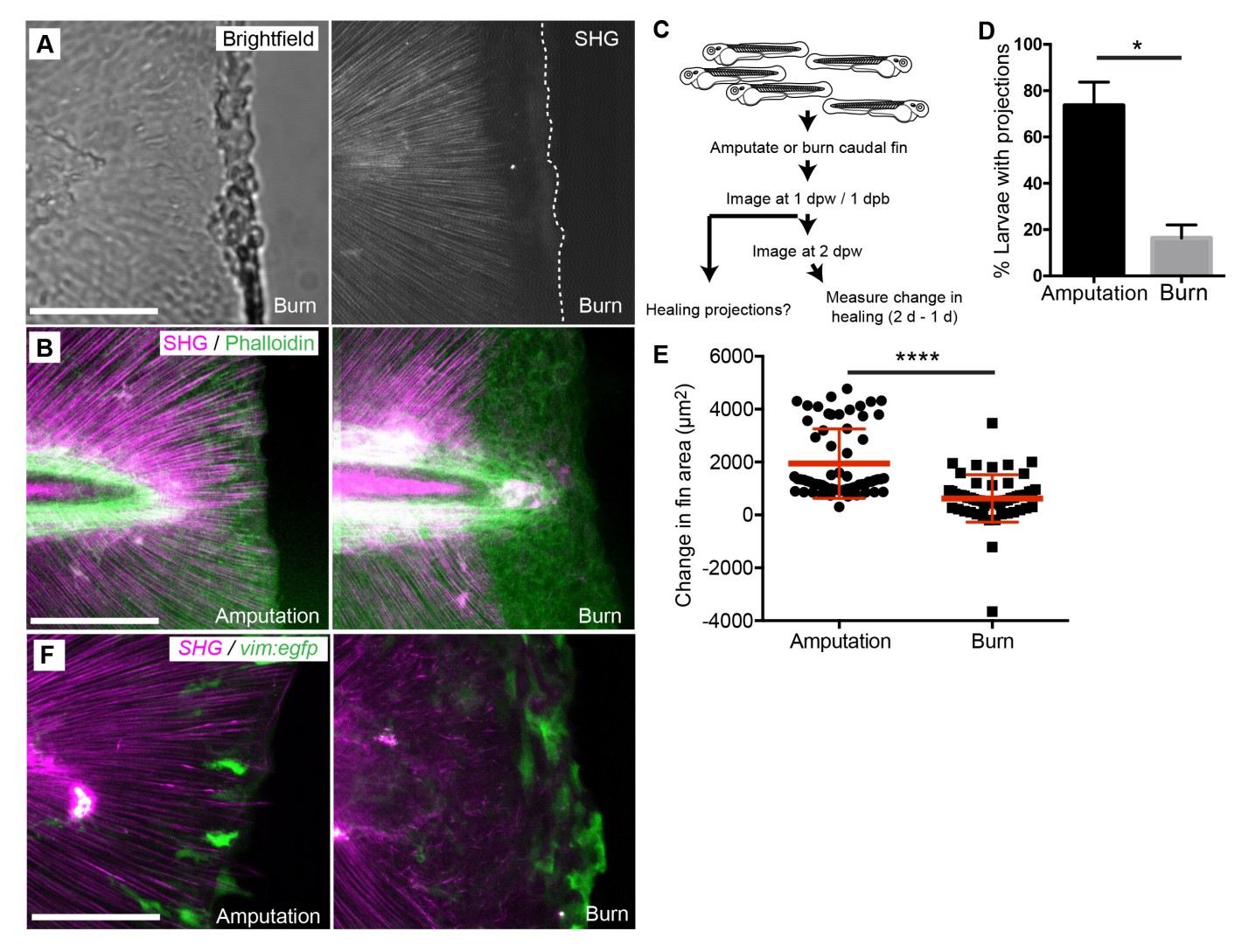

**Figure 7.** Burn wounding impaired the formation of epithelial and collagen projections and wound healing. (**A**) Burning of the caudal fin resulted in a striking absence of SHG fibers near the wound edge. (**B**) In contrast to the amputation assay, a large gap between the edge of the burned fin and the edge of the intact SHG fibers was observed at 24 hpw using a combination of SHG imaging and phalloidin staining, observed in two experiments. (**C**) Schematic of burn wound experiments and analysis. (**D**) Burn wounds displayed a significant reduction in epithelial projections at 24 hpw compared to transection wounds (p=0.0116; n = 3 replicate percentages: for amputation 108 larvae total and for burn 62 larvae total were scored over the three replicates, with 17 to 24 larvae per treatment per replicate; amputation 95% CI = 48.66% to 98.67%, burn 95% Ci = 2.21% to 30.46%). (**E**) Healing area was significantly reduced in the burn wounded larvae compared to amputated fins (p<0.0001, amputation n = 65 larvae, burn n = 58 larvae over three replicates, with 16 to 22 larvae per treatment per replicate; amputation 95% CI = 1750.47 to 2166.58, burn 95% CI = 497.69 to 939.76). (**F**) vim+ cells are present at the wound edge at 24 hpb, but the morphology is different compared to tail transection, as observed in two experiments. Scale bars represent 100 μm. *p<0.05, ****p<0.0001, error bars are standard deviation.

DOI: https://doi.org/10.7554/eLife.30703.038

The following source data and figure supplement are available for figure 7:

**Source data 1.** Proportion of larvae exibiting visible projections with amputation or burn wound.
DOI: https://doi.org/10.7554/eLife.30703.040
**Source data 2.** Change in caudal fin regenerate area with amputation or burn wound.
DOI: https://doi.org/10.7554/eLife.30703.041
**Figure supplement 1.** Collagen disruption impaired regeneration following amputation of the caudal fin.
DOI: https://doi.org/10.7554/eLife.30703.039

larvae (*Figure 7C*). Compared to amputated fins, burn wounds resulting in a loss of SHG fiber integrity led to a significant reduction in the percentage of larvae displaying epithelial projections at 24 hpb (*Figure 7D*, *Figure 7—source data 1*). Furthermore, when we assessed the wound healing (length and area) of individual larvae between 24 and 48 hr post burn or post amputation, the burn wound had significant defects in healing as compared to tail transection (*Figure 7E*, *Figure 7—source data 2*, *Figure 7—figure supplement 1G*). Interestingly, vim+ cells also appeared at the wound edge in the burn wound, although the morphology of the cells were different compared to tail transection (*Figure 7F*). It is particularly intriguing that a mechanical mode of collagen disruption using the burn model had such a dramatic effect on collagen reorganization after injury and subsequent regeneration. Taken together, these data suggest that the formation of collagen projections at the wound edge is necessary for the formation of epithelial extensions and efficient wound repair.

## Discussion

In this study, we demonstrated that vimentin functions downstream of wound-induced redox signaling and regulates collagen expression and reorganization during wound repair. Using live imaging, our work uncovered a previously unknown mechanism whereby collagen projections at the wound edge guide epithelial growth during fin regeneration, a process dependent on vimentin. In response to wounding, cells at the wound edge turn on *vimentin* expression and are critical for an increase in *collagen* expression and subsequent collagen reorganization at the wound that promotes proper tissue repair. This finding suggests that vim+ cells affect the behavior of adjacent tissues by regulating the scaffold needed for regenerative outgrowth through the polarized formation of epithelial membrane extensions. Further, direct disruption of collagen organization by burn wound impairs the formation of the projections and subsequent regrowth.

Our findings suggest that vim+ cells are critical to generate a wound microenvironment that optimizes for efficient wound healing in zebrafish larvae. It was an unexpected finding that the vim+ cells in the fin did not migrate to the wound margin, but rather remained stationary while a population of cells at the wound turned on expression of *vimentin*. This population of cells was also not particularly motile but appeared to be critical for regulating the collagen scaffold for epithelial projections that ultimately led to efficient wound repair. The idea that vimentin regulates cell migration has been long established, however, recent studies have also highlighted the role of vimentin in regulating ECM production (*Cheng et al., 2016*; *Darby et al., 2014*; *Eckes et al., 2000*). Vimentin regulates collagen mRNA through Larp6 (*Challa and Stefanovic, 2011*) and was also shown to regulate collagen accumulation at the wound in mouse models (*Cheng et al., 2016*). Herein, we support, and extend, the evidence of these previous studies by showing a critical role for vimentin in both collagen expression and its dynamic reorganization during the repair process in zebrafish larvae.

Substantial evidence implicates injury-induced ROS as a key player in modulating tissue regeneration. Prior studies indicate ROS regulates growth factor pathways such as Wnt/fgf signaling after amputation in Xenopus tadpoles (*Love et al., 2013*). Our findings identify a new pathway through which ROS can regulate regeneration through a population of vim+ cells at the wound margin. We propose that ROS stimulates *vimentin* expression in a population of cells at the wound through its effects on NFκB activity. NFκB is known to be a redox sensitive transcription factor (*Nakajima and Kitamura, 2013*). Moreover, wound induced activation of DUOX1 and the generation of hydrogen peroxide appears to be critical for the activation of NFκB reporter activity in zebrafish larvae (*de Oliveira et al., 2014*). Taken together, these findings provide a link between ROS, NFκB and the organization of collagen at the wound margin to promote healing. Moreover, these findings, in combination with published work, suggest that redox signaling may have beneficial effects on the ECM in the wound microenvironment to promote wound healing in zebrafish. It is interesting to speculate that this may also be the case in humans, rather than just having the detrimental effects suggested by some clinical studies (*Dhall et al., 2014*).

Long-term time-lapse imaging of SHG and epithelial cell dynamics demonstrated a close association between the extending SHG filaments and the epithelial cell projections. This imaging supports a role for these collagen projections in guiding the epithelial membrane extensions. As the collagen projections progressed the epithelial cells showed close apposition to these ECM structures. In future studies, it will be intriguing to investigate the adhesion sites that form in these projections and if integrin-containing focal contact sites are critical for this guidance. The absence of projections

when collagen cross-linking was impaired supports a key role for these structures in tissue guidance. A caveat is that this treatment was ultimately toxic at later developmental stages, preventing longer term analysis of these effects.

Further support for a role of these SHG fibers in tissue guidance and wound healing was based on our findings with caudal fin burn wounds. The localized caudal fin burn wound led to a complete loss in SHG signal in the wound margin. This burn wound response was surprisingly different from the wound ECM response in the well-studied tail transection model and makes it a unique model for future studies. This type of thermal damage abrogated epithelial projections and significantly impaired regeneration and repair of the fin in the time frame examined. It is also relevant to highlight that fin growth during normal development does not exhibit this type of collagen guidance, but rather progresses as a smooth surface without projections. These findings highlight the key role for collagen as a guidance cue that can promote wound healing in some contexts. In agreement with this idea substantial clinical efforts have focused on utilizing collagen scaffoldings with promising results (*Kim et al., 2016*; *Reilly et al., 2017*; *Tracy et al., 2016*).

Taken together, our findings suggest that caudal fin transection induces a ROS mediated program leading to a collagen matrix organization that guides epithelial behavior and wound repair. Key to this work was the ability to live image the process of wound healing over extended durations. Using a new reporter of vim+ mesenchymal cells our findings revealed that this cell population did not migrate during wound healing but rather regulated the wound microenvironment to promote efficient regeneration. To our knowledge, this was the first time that collagen structure using SHG and epithelial cell membranes were imaged long term in vivo, allowing the identification of these dynamic cell projections that occur during fin growth after injury. This type of study highlights the power of zebrafish for imaging dynamic processes over time in vivo to uncover new mechanisms of regulation. However, a caveat is that it remains unclear how these observations translate to mammalian wound healing. As cutaneous wounds are a worldwide clinical problem, the findings suggest the possibility that zebrafish may be a powerful tool to identify new mechanisms that regulate collagen re-organization, guiding future development of therapeutics focused on optimal repair of different types of wounds, including mechanical injury and burn wounds.

# Materials and methods

## Key resources table

| Reagent type (species) or resource | Designation | Source or reference | Identifiers | Additional information |
|---|---|---|---|---|
| Genetic reagent (Danio Rerio) | vimentin(vim) | this paper | | crispr RNA targeting exon 1 (5'- CAAAGGAGCGTCCCGGGGT – 3') |
| Strain (D. Rerio) | Tg(−2vim:EGFP) | this paper | | |
| Chemical compound | DPI | Tocris | | 100 uM |
| Chemical compound | BAPN | Sigma | | 140 uM |
| Chemical compound | Withferin A (WA) | Santa Cruz | | 30 uM |
| Sequence-based reagent (D.Rerio) | duox | *Niethammer et al. (2009)* | | MO |
| Sequence-based reagent (D.Rerio) | p53 | | | MO |
| Sequence-based reagent (D.Rerio) | vim1 | this paper | | MO 80 uM GTAATAGTGCCAGAACAGACCTTCTC |
| Sequence-based reagent (D.Rerio) | vim2 | this paper | | MO 110 uM TCTTGAAGTCTGGAAATGAGATGCA |
| Sequence-based reagent (D.Rerio) | larp6 | | | MO 80 uM GGGTGTTGGTCTTACCTTCTTGAA |
| Gene(D. Rerio) | vimentin | ENSDARG00000010008 | | |
| Actin label | phalloidin | Invitrogen | | 1:100 |
| Antibody | col2a | PAb, II-II6B3-s; DHSB | | 1:200 |
| Strain (D. Rerio) | krt4:EGFP-CAAX | *Krens et al. (2011)* | | |
| Strain (D. Rerio) | krt4:TdTomato | *Yoo et al. (2010)* | | |

*Continued on next page*

Continued

| Reagent type (species) or resource | Designation | Source or reference | Identifiers | Additional information |
|---|---|---|---|---|
| Sequence-based reagent (D.Rerio) | vimentin | this paper | | ISH probes<br>F: CTTCAACAATAACCCGCAAA<br>R:TAATACGACTCACTATAGGGGG<br>TCAGGTTTGGTCACTTCC |
| Sequence-based reagent (D.Rerio) | col1a1 | this paper | | RT primers<br>F:TGTCACTGAGGATGGTTGCAC<br>R:GCAGACGGGATGTTTTCGTTG |
| Sequence-based reagent (D.Rerio) | col2a1b | *Durán et al. (2011)* | | RT primers<br>F:AACAGAAGTGCTTCCGAACG<br>R:TGCTCTGGTTTCTCCCTCAT |
| Sequence-based reagent (D.Rerio) | vimentin | this paper | | RT primers<br>F:ACCGGGGAAAAGAGCAAAGT<br>R:CGAGCCAGAGAGGCGTTATC |
| Sequence-based reagent (D.Rerio) | larp6 | this paper | | RT primers<br>F:CAAACTGGGCTTCGTCAGTG<br>R:TCCGTTGTTGGAATCTCCGC |
| Recombinant DNA reagent(D.Rerio) | tol2:krt4-hras-g(v)12-rfp | *Freisinger and Huttenlocher (2014)* | | |
| Software | CurveAlign | *Liu et al., 2017* | | https://loci.wisc.edu/<br>software/curvealign |
| Software | FIJI, ImageJ | *Schindelin et al. (2012)*; | | https://fiji.sc/ |
| Software | Imaris | Bitplane | | |

## Zebrafish maintenance and handling

All protocols using zebrafish in this study were approved by the University of Wisconsin-Madison Research Animals Resource Center. Adult zebrafish and embryos were maintained as described previously (*Yoo et al., 2010*). For wounding assays, 2 or 3 day post-fertilization (dpf) larvae were anesthetized in E3 medium containing 0.2 mg/ml Tricaine (ethyl 3-aminobenzoate; Sigma-Aldrich). To prevent pigment formation, some larvae were maintained in E3 medium containing 0.2 mM *N*-phenylthiourea (Sigma-Aldrich). Adult AB strain fish, including transgenic zebrafish lines, *Tg(krt4:egfp-caax)* (*Krens et al., 2011*), *Tg(mpeg1:dendra2)* (*Harvie et al., 2013*), *Tg(krt4:tdtom)*, and the *casper* mutant line (*White et al., 2008*) were utilized.

## Generation of *Tg(−2vim:egfp)* transgenic zebrafish

The 2 kb upstream region of the putative translational start site of the zebrafish vimentin gene was PCR amplified from BAC CH211-48N12 (BACPAC) using the following primers:

XhoI-Forward Primer (5'-GAT<u>CTCGAG</u>TGTTGCCGTACGTTATTTGC-3')

KpnI-Reverse Primer (5'-GAT<u>GGTACC</u>CTAAATATCGCACCTGTCCA-3')

The resulting 2 kb PCR product was gel purified, sequentially digested (XhoI and KpnI) and cloned into an expression vector containing EGFP, minimal Tol2 elements for efficient integration (*Urasaki et al., 2006*) and an SV40 polyadenylation sequence (Clontech Laboratories, Inc.). F0 larvae were obtained by injecting 3 nL of 12.5 ng/µL DNA plasmid and 17.5 ng/µL in vitro transcribed (Ambion) transposase mRNA into the cytoplasm of one-cell stage embryo. F0 larvae were raised to breeding age and crossed to adult AB zebrafish. F2 founders were screened for EGFP expression using a Zeiss Axio Zoom stereo microscope (EMS3/SyCoP3; Zeiss; Plan-NeoFluar Z objective).

## *Tg(−2vim:egfp)* expression with EMT induction (HRas$^{G12}$ and HRas$^{V12}$ overexpression)

Either *tol2:krt4-hras-g12-rfp* or *tol2:krt4-hras-v12-rfp* was injected into *Tg(−2vim:egfp)* embryos at the one cell stage (*Freisinger and Huttenlocher, 2014*). At 2 dpf, fluorescence images were acquired with a confocal microscope (FluoView FV1000; Olympus) using a NA 0.75/20x objective. Each fluorescence channel was acquired by sequential line scanning. Z series were acquired using 180–280 µm pinhole and 0.5–5 µm step sizes. Z series images were stacked or 3D reconstructed by the FluoView FV1000 software.

## Whole mount In situ hybridization

For whole mount in situ hybridization, larvae were fixed in 4% paraformaldehyde in PBS and mRNA was labeled by in situ hybridization as previously described (*Long and Rebagliati, 2002*). In short, both Dig-labeled antisense probes were hybridized using a 55°C hybridization temperature. Purple color was developed with AP-conjugated anti-DIG and BM purple (Roche Applied Science). Reactions were stopped in PBS. Imaging was performed with a Nikon SMZ-1500 stereoscopic zoom microscope. The T7 promoter was attached 3' of the coding sequence of primers to make the DNA template for the vimentin probe. After sequence confirmation of the DNA template, labeled RNA was transcribed with the use of T7 RNA polymerase (Ambion).

Oligo sequences used for PCR were as follows:
Vimentin F: 5'-CTTCAACAATAACCCGCAAA-3'
T7 Vimentin R: 5'-TAATACGACTCACTATAGGGGGTCAGGTTTGGTCACTTCC-3'
T7 Vimentin Sense F: 5'-TAATACGACTCACTATAGGGCTTCAACAATAACCCGCAAA-3'
Vimentin Sense R: 5'-GGTCAGGTTTGGTCACTTCC-3'

## Regeneration assays and drug treatments

For regeneration assays, tail transection was performed on 2–2.5 dpf larvae using a surgical blade (Feather, no. 10). Regenerate length was quantified by measuring the distance between the caudal tip of the notochord and the caudal edge of the tail fin at 3 days post-wounding (dpw). Regenerate area was measured using the FIJI (*Schindelin et al., 2012*) image analysis software to assess the total fin area posterior to the notochord. Diphenyleneiodonium (DPI) (Tocris) (100 µM) or Withaferin A (WA) (Santa Cruz) (30 µM) was applied for 1 hr before and after wounding. β-aminopropionitrile (BAPN) (Sigma Aldrich), at a concentration of 140 µM, was maintained from the time of wounding through the duration of the healing process assessed (2 dpw). The treatment was replenished at 24 hpw. Larvae were imaged at 2 or 3 dpw for regeneration studies on a Zeiss Zoomscope (EMS3/SyCoP3; Zeiss; Plan-NeoFluar Z objective).

## Live imaging in zWEDGI device

To minimize larval movement without impinging on caudal wound healing, larvae were mounted in a zWEDGI device, as previously described (*Huemer et al., 2017*). Briefly, an anesthetized larva was loaded into the zWEDGI chamber such that the head remained in the loading chamber while the tail passed through the restraining tunnel and protruded into the wounding chamber. 1% low melting point agarose (Sigma-Aldrich) in Tricaine/E3 was placed over the larva's head, filling the loading chamber and allowed to solidify with the larva in the proper position. Additional Tricaine/E3 was added as needed. The tail, freely suspended in the Tricaine/E3 could then be imaged through the cover glass bottomed dish, using either confocal or multiphoton SHG microscopy (described below). For time-lapse imaging of projection formation, larvae were imaged every 15–20 min (depending on slice number) with a confocal microscope (FluoView FV1000; Olympus) using a NA 0.75/20x objective from 20 mpw through 48 hpw. For time-lapse imaging of *Tg(−2vim:egfp)* following amputation, images were acquired every 15–25 min for up to 70 hpw with a confocal microscope (FluoView FV1000; Olympus) using a 20X NA 0.75 objective. At roughly 48 hpw, larvae were removed from the zWEDGI device and re-inserted to maintain viability. Images were the processed into videos using FIJI.

## Quantification of EGFP intensity

To assess EGFP expression in the *Tg(−2vim:egfp)* line following amputation, quantification of EGFP expression at the wound edge was performed in FIJI on Sum intensity z-projections. The mean fluorescence intensity was measured in 100 micron long by 30 micron wide rectangular region aligned along the wound edge and centered at the level of the notochord. In the same fin, as an assessment of the background fluorescence, a 40 micron diameter circle was used to measured in an area of the fin anterior to the wound edge and devoid of obvious cellular EGFP expression. The ratio of the mean intensity at the wound edge to the mean intensity of the background in the fin was used as an indicator of the proportional signal intensity of the wound edge above background.

## Immunofluorescence and phalloidin label

Larvae at 2.5–3.5 dpf were fixed with 1.5% paraformaldehyde in 0.1 M Pipes (Sigma-Aldrich), 1.0 mM MgSO$_4$ (Sigma-Aldrich), and 2 mM EGTA (Sigma-Aldrich) overnight at 4°C and immunolabeled as previously described (Yoo and Huttenlocher, 2011). Mouse anti-collagen type II (PAb, II-II6B3-s; DHSB) was used at 1:200 in PBS. Dylight 488-conjugated donkey anti-mouse IgG antibodies (Jackson ImmunoResearch Laboratories, Inc.) were used as secondary antibodies at 1:250. Immunofluorescence images were acquired with a confocal microscope (FluoView FV1000; Olympus) using a NA 0.75/20x objective. Each fluorescence channel was acquired by sequential line scanning. Z series were acquired using 180–280 μm pinhole and 0.5–5 μm step sizes. Z series images were stacked or 3D reconstructed by the FluoView FV1000 software. For phalloidin staining of actin, larvae were fixed in 4% PFA for a minimum of 4 hr at RT or O/N at 4°C. Larvae were washed three times in PBS-Tween20 (0.1%) and permeabilized for 1 hr at RT in PBS-Triton (2%). Larvae were then incubated O/N at 4°C in rhodamine-Phalloidin (Invitrogen) or Alexa Fluor 488-Phalloidin (Invitrogen) diluted 1 to 100 in PBS-Triton (2%). Imaging was performed after 3X washes in PBS-Tween (0.1%).

## Multiphoton microscopy of second harmonic generation (SHG), collagen type two antibody label and phalloidin label

Fixed caudal fin samples were prepared as previously described (LeBert et al., 2016; LeBert et al., 2015) and imaged on a custom-built multiphoton microscope (Conklin et al., 2011; LeBert et al., 2016) at the Laboratory for Optical and Computational Instrumentation using either a 40X long working distance water immersion lens (1.2 NA, Nikon) for projection analyses or a 60X VC water immersion lens (1.20 NA, Nikon) for antibody and phalloidin label experiments, with the laser (Coherent Chameleon) tuned to 890 nm. Backwards SHG was collected using a 445/20 nm emission filter (Semrock, Rochester NY) while the fluorescent signal from the collagen type two antibody label was sequentially collected using a 520/35 nm emission filter (Semrock) and both signals were detected using a H7422P-40 GaAsP Photomultiplier Tube (PMT) (Hamamatsu, Japan). Rhodamine-phalloidin label was collected sequentially with the SHG images using a 542/27 nm bandpass emission filter (Semrock). Bright field images were simultaneously collected using a separate photodiode based transmission detector (Bio-Rad, Hercules CA). Imaging parameters remained constant across imaging days for a given experiment. For projection analyses data were collected as z-stacks with optical sections two microns apart, at 512 × 512 resolution. For SHG with collagen type two antibody label, images were collected as z-stacks with optical sections one micron apart, at 1024 × 1024 resolution. For SHG with phalloidin label, images were collected as z-stacks with optical sections one micron apart, at 512 × 512 resolution. Z projections were generated using FIJI software while 3D reconstructions and surface renderings were constructed using Imaris software (Bitplane, Zurich, Switzerland).

## Time-Lapse multiphoton microscopy of projections

To observe the dynamic interaction of SHG projection and epithelial cells, either *casper* mutants (White et al., 2008) or *Tg(krt4:egfp-caax)*(Krens et al., 2011) larvae were wounded at 2 dpf and mounted into zWEDGI device at 20 to 25 hpw. The wound region was imaged using multiphoton microscopy, as described above. Images were collected using a 20X VC air objective (NA = 0.75, Nikon) as z-stacks, two micron optical sections, approximately every 15 min for 20 to 23 hr at 512 × 512 resolution, with 890 nm excitation. SHG and bright field images were collected simultaneously while the SHG and GFP z-stacks were collected sequentially, with the GFP emission collected using a 520/35 emission filter (Semrock). Z projections were generated using FIJI software while 3D reconstructions and surface renderings were constructed using Imaris software (Bitplane, Zurich, Switzerland).

## LC-PolScope microscopy

Fixed wounded caudal fin samples were prepared in a similar method as described above for SHG microscopy in that tails were removed from the body and mounted in PBS in a glass bottomed imaging dish. For LC-PolScope (Oldenbourg, 2005) imaging, a glass coverslip was placed over the opening of the imaging dish, allowing the condenser to approach the sample without contacting the mounting fluid while not compressing the sample. Images were collected using either a 20x

(NA = 0.4) or a 40X (NA = 0.95) air objective (Nikon) on a Nikon Eclipse TE200 microscope with 549/15 nm interference filter and a dry condenser (NA = 0.85). The camera was an ORCA-Flash4.0 V2 digital CMOS (Hamamatsu, Japan). The system was controlled using the OpenPolScope hardware and software for birefringence imaging (openpolscope.org) (*Keikhosravi et al., 2017*).

## Analysis of projections at the wound edge

Larvae were fixed at 1 dpw (3 dpf) as described above. Caudal fins were imaged using a Zeiss Zoomscope microscope using a Zeiss PlanNeoFluar Z 1X:0.25 FWD 56 mm lens. Images were then assessed for the presence or absence of projections along the amputated plane.

## Analysis of SHG projections at the wound edge

Because of the three dimensional nature of the caudal fin, for ease of analysis sum z-projections of SHG images were generated in FIJI (*Schindelin et al., 2012*). In FIJI, a 512 pixel (215 micron) region, at the wound edge distal to, and centered on, the tip of the notochord was used as the region of interest (ROI). Contour length was determined by a freehand line tracing the detailed contour of the fibers at the wound edge. Analysis of SHG fiber angle relative to the wound edge was conducted using CurveAlign open source software (*Bredfeldt et al., 2014*); liu et al., 2007; *Schneider et al., 2013*). SHG and bright field z-stacks were z-projected using FIJI (https://fiji.sc/), and the resulting 2D images were used for analysis. Within the CurveAlign software, the wound edge was drawn on the bright field image and then applied as a boundary for the CurveAlign analysis. In CurveAlign, curvelets fiber representation (CFR) mode was used to track individual fibers and each fiber was represented as a group of curvelets that have localized fiber orientation information. The angle of each curvelet, relative to the wound boundary and within a 60 micron distance from that boundary, was determined. The mean value of these angles was used as the overall orientation value for each tail.

## RNA extraction for qRT-PCR

RNA was extracted from an approximately 500 μm length portion of the fin using the miRvana RNA purification kit (Ambion). First strand cDNA synthesis was performed using super script III (Invitrogen). Resulting cDNA was diluted 1:10 in RNAse free water before qRT-PCR was performed in a minimum of triplicate using Roche green master mix (Roche) for *rps11*, *col1a1* and *col2a2* from purified fin RNA. Fold change was determined using efficiency-corrected comparative quantitation. Data were normalized to no wound age matched control samples. Primers:

Rps11:
F-5'-TAAGAAATGCCCCTTCACTG-3' R-5'-GTCTCTTCTCAAAACGGTTG-3'
Col1a1:
F-5'-TGTCACTGAGGATGGTTGCAC-3' R-5'-GCAGACGGGATGTTTTCGTTG-3'
Col2a1b (*Durán et al., 2011*):
F- 5'-AACAGAAGTGCTTCCGAACG-3' R- 5'-TGCTCTGGTTTCTCCCTCAT-3'

## Morpholino and RNA injections

Morpholino oligonucleotides (Gene Tools) were re-suspended in water to a stock concentration of 0.5 mM. Final morpholino concentrations were injected into 1–2 cell stage embryos in 3 nl amounts and embryos were maintained at 28.5°C. The DUOX morpholino was used as published (*Yoo et al., 2012*). The *vimentin* and *Larp6* splice blocking morpholinos used were as follows:

*vim* MO1 targeting Exon 1- Intron 1 (80 μm):
5'-GTAATAGTGCCAGAACAGACCTTCTC-3'
*vim* MO2 targeting Intron 1-Exon 2 (110 μm):
5'-TCTTGAAGTCTGGAAATGAGATGCA-3'
*larp6* MO targeting Exon 2-Intron 2 (80 μm):
5'-GGGTGTTGGTCTTACCTTCTTGAA-3' control MO
5'-CCTCTTACCTCAGTTACAATTTATA-3'

To analyze knockdowns, RT-PCR was performed on RNA isolated from single larvae at 2 dpf. The following primers were used to assess knockdown:
Vim MO 1 and 2:
F-5'-ACCGGGGAAAAGAGCAAAGT-3' R-5'-CGAGCCAGAGAGGCGTTATC-3'

Larp6 MO:
F-5'-CAAACTGGGCTTCGTCAGTG-3' R-5'-TCCGTTGTTGGAATCTCCGC-3'

Rescue experiments were performed using zebrafish *vim* RNA. In short, a gene block (IDT) for vimentin was used as the template. The sequence was cloned into the PCS2 +vector from which in vitro transcription was performed (Ambion). RNA was purified with the miRvana RNA purification kit (Ambion). RNA was injected at 125 ng/µl at the 1–2 cell stage in 3 nl amounts.

## Whole mount terminal deoxynucleotidyl transferased UTP Nick-End labeling (TUNEL)

In brief, caudal fins from 2 dpf larvae were amputated and larvae were fixed in 4% PFA-PBS at 4°C overnight. The larvae were washed three times in PBS and stored overnight at −20°C in MeOH. The TUNEL labeling was performed as instructed by the manufacturer (Roche). Analysis was performed using a Zeiss Zoomscope. Label bleaches quickly, so acquiring images was not always possible.

## Vimentin CRISPR-Cas9

CRISPR guide RNA was designed using crisprscan: http://www.crisprscan.org

Exon 1 target sequence: 5'- CAAAGGAGCGTCCCGGGT – 3'

The pT7 gRNA vector (Addgene 46759) was digested with BsmBI, BglII and SalI (New England Biolabs) and diluted to 5 ng/µl in ddH$_2$O. Annealed oligos were ligated into the vector using quick ligase (New England Biolabs). Single colonies were selected following transformation and digest confirmed with BglII. Candidate plasmids were sequenced using M13 primer. Sequence confirmed plasmids were linearized with BamHI (New England Biolabs) and in vitro transcribed using MAXI-Script T7 kit (Ambion, Life Technologies). The resulting guide RNA was injected into the yolk at the one cell stage at a volume of 2 nl. The final concentrations of the injection mixes were as follows: gRNA at ~40 ng/µl and Cas9 protein (New England Biolabs) at ~55 ng/µL. To confirm, 2–5 dpf larval zebrafish were individually digested overnight at room temperature in 100 µl DNAzol plus 200 µg/µl proteinase K. PCR was performed on gDNA, diluted 1:10 to confirm CRISPR cuts with the following primers for *vim1c*:

Vim1c Forward: 5' – GACAAAGTGCGCTTTCTGGA–3'
Vim1c Reverse: 5' – TCCACCTCCACTTTGCTCTT–3'

The F0 mosaic larvae amplicon was then TOPO-cloned and sequenced to verify the presence of mutations. Clutches containing larvae positive for CRISPR cuts were grown to adulthood. At breeding age, sperm or eggs were collected and gDNA was isolated as described above. The target sequence was amplified by PCR and analyzed by Indel Detection by Amplicon Analysis (IDAA) as previously described (*Yang et al., 2015*). Samples were analyzed using PeakScanner2.0 (Applied Biosystems). Potential heterozygous mutants were then TOPO-cloned and sequenced to determine the germline mutation. PCR primers for IDAA PCR amplification were as follows:

FAM Vim1c Forward:
5'-GTAAAACGACGGCCAGTGGCGCACGTCCTACAACTACA-3'
Vim1c Reverse: 5'-TCGATGTCCTCCCCTAGGTT-3'

## Caudal fin burn

At 2 dpf zebrafish were placed into a 60 mm dish in E3 +0.2 mg/ml Tricaine. A fine tip cautery (BVI Accu-Temp; cat #8442000) was used to burn the caudal fin. The cautery was turned on after being placed into the water and was held on the posterior tip of the caudal fin for 1–2 s, until a slight bend in the fin was apparent. After the burn, fish were placed into a clean milk-treated 35 mm petri dish with fresh E3 and held in an incubator at 28.5 degrees. For regeneration assessment, embryos were imaged at 1 and 2 dpw on a Zoomscope (EMS3/SyCoP3; Zeiss; Plan-NeoFluar Z objective). Images were taken on an Axiocam MRm CCD camera using ZEN pro 2012 software (Zeiss).

## Statistical analyses

Statistical parameters are included in the figure legends including the number of experimental replicates and total 'n'. For projection analyses or representative single experiment graphs, assuming a Gaussian distribution of the overall population of values, p-values were determined by two-tailed paired *t* test (two comparisons) or one-way analysis of variance (ANOVA) (multiple comparisons)

comparing means of each sample (GraphPad Prism, GraphPad Software, San Diego CA). For analysis of experiments consisting of values from multiple replicates, such as the contour length and fiber orientation measurements, as well as relative intensity and regenerate area and length measurements, Least Squared Means analysis in R (www.r-project.org) (*Vincent et al., 2016*) was performed on multiple replicate experiments, using Tukey method when comparing more than two treatments.

## Acknowledgements

AH was funded by NIH R35 GM1 18027 01, NG by Molecular Biosciences Training Grant T32-GM07215 and Laboratory for Optical and Computational Instrumentation and the Morgridge Institute for Research (KE). We thank Yuming Liu and Adib Keikhosravi for their assistance with the CurveAlign analysis and PolScope microscopy, respectively.

## Additional information

### Funding

| Funder | Grant reference number | Author |
| --- | --- | --- |
| National Institute of General Medical Sciences | GM1 18027 01 | Anna Huttenlocher |

The funders had no role in study design, data collection and interpretation, or the decision to submit the work for publication.

### Author contributions

Danny LeBert, Conceptualization, Resources, Data curation, Software, Formal analysis, Methodology, Writing—original draft; Jayne M Squirrell, Conceptualization, Data curation, Software, Formal analysis, Supervision, Validation, Visualization, Methodology, Writing—original draft; Chrissy Freisinger, Conceptualization, Data curation, Methodology, Writing—review and editing; Julie Rindy, Data curation, Formal analysis, Investigation, Methodology; Netta Golenberg, Data curation, Formal analysis, Investigation, Writing—review and editing; Grace Frecentese, Data curation, Investigation; Angela Gibson, Conceptualization, Formal analysis, Writing—review and editing; Kevin W Eliceiri, Conceptualization, Funding acquisition, Investigation, Writing—review and editing; Anna Huttenlocher, Conceptualization, Formal analysis, Supervision, Funding acquisition, Investigation, Methodology, Writing—original draft, Project administration

### Author ORCIDs

Danny LeBert (iD) http://orcid.org/0000-0002-3729-2583
Jayne M Squirrell (iD) http://orcid.org/0000-0003-0651-6027
Kevin W Eliceiri (iD) http://orcid.org/0000-0001-8678-670X
Anna Huttenlocher (iD) http://orcid.org/0000-0001-7940-6254

### Ethics

Animal experimentation: This study was performed in strict accordance with the recommendations in the Guide for the Care and Use of Laboratory Animals of the National Institutes of Health. All of the animals were handled according to approved institutional animal care and use committee (IACUC) protocols (#08-133) of the University of Wisconsin.

### Decision letter and Author response

Decision letter https://doi.org/10.7554/eLife.30703.044
Author response https://doi.org/10.7554/eLife.30703.045

## Additional files

### Supplementary files

• Transparent reporting form
DOI: https://doi.org/10.7554/eLife.30703.042

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
