## [Decision Letter]

Thank you for submitting your article "Damage-induced reactive oxygen species regulates *vimentin* and dynamic collagen-based projections to mediate wound repair" for consideration by *eLife*. Your article has been reviewed by two peer reviewers, and the evaluation has been overseen by Fiona Watt as the Senior and Reviewing Editor. The following individual involved in review of your submission has agreed to reveal his identity: Victoriano Mulero (Reviewer #1).

The reviewers have discussed the reviews with one another and the Reviewing Editor has drafted this decision to help you prepare a revised submission.

Summary:

This study reports the generation of a *vimentin* reporter zebrafish line to study in vivo, in real time, EMT during wound repair. Pharmacological inhibition studies show that *vimentin* activation in the wound is downstream of ROS and NFkB. Genetic experiments show that *vimentin* is required for regeneration and vim+ cells are associated with, and required for, extension of collagen fibers. Finally, disruption of collagen fibers by impairing their crosslinking or in a new burn-wounding model impairs healing. The study is well performed and provides new insights into the role of *vimentin* in the healing process.

Major points:

1) The morpholino activity is not sufficiently validated in comparison with the CRISPR-Cas9 mutant. The reviewers are not suggesting that the authors should repeat all the experiments using the mutant; however, additional validation of the morpholino effects would strengthen the work significantly.

2) Figure 2 and Figure 2—figure supplement 1: although the pictures and Video 1 show that epithelial (*krt4*+) and mesenchymal (vim+) cells do not colocalize in the wound edge, the video should be performed in *krt4:tom;vim:gfp* larvae to confirm that keratinocytes do not show EMT. In addition, the authors should use a genetic inhibition approach, at least for ROS, to confirm the results, since DPI is not specific and side effects are expected at the high concentration used. Imaging of ROS and NFkB using appropriate reporters may also reveal if these signals are intrinsically required by cells activating *vimentin* in the wound.

3) Figure 4, Figure 6, Figure 4—figure supplement 1, Figure 5—figure supplement 1 and Figure 6—figure supplement 1: as in point 2, genetic inhibition and reporters would be very informative.

4) Figure 7: these results show that collagen fiber disruption impairs regeneration. However, the link between these results and the main finding of the study, i.e. the role of *vimentin* in wound healing, is not clear. It would be valuable to use the *vimentin* reporter here.

---

## [Author Response]

Major points:1) The morpholino activity is not sufficiently validated in comparison with the CRISPR-Cas9 mutant. The reviewers are not suggesting that the authors should repeat all the experiments using the mutant; however, additional validation of the morpholino effects would strengthen the work significantly.

We now provide additional information about the target region for the CRISPR in the manuscript (Figure 3). We have also shown that transient CRIPSR replicates the findings shown with MO (Figure 3). It is important to note that the *vimentin* deficient line recapitulates findings in the *vimentin* knockout mice where it has been previously reported that there is a defect in wound healing and collagen accumulation at the wound. Our findings extend these reports by performing live imaging of wound healing using a novel *vimentin* reporter line. We spent substantial effort to develop and characterize the *vimentin* mutant; unfortunately, the mutant is not ready for validation and will require several more outcrosses before it can be used for these studies.

2) Figure 2 and Figure 2—figure supplement 1: although the pictures and Video 1 show that epithelial (krt4+) and mesenchymal (vim+) cells do not colocalize in the wound edge, the video should be performed in krt4:tom;vim:gfp larvae to confirm that keratinocytes do not show EMT. In addition, the authors should use a genetic inhibition approach, at least for ROS, to confirm the results, since DPI is not specific and side effects are expected at the high concentration used. Imaging of ROS and NFkB using appropriate reporters may also reveal if these signals are intrinsically required by cells activating vimentin in the wound.

We have performed live imaging and do not see co-localization of Krt4 and *vimentin*. This does not rule out an EMT event however. We include a new panel and video to address this concern (Figure 2—figure supplement 1 (panel C). We have revised the text accordingly.

We also include new supplemental data using Duox MO that shows that Duox regulates *vimentin* reporter activity at the wound (Figure 2—figure supplement 1 (panels F, G). In addition, we now provide quantification from 3 experiments of fluorescence intensity measurements at the wound edge in response to the DPI treatment from 3 separate treatments (Figure 2).

3) Figure 4, Figure 6, Figure 4—figure supplement 1, Figure 5—figure supplement 1 and Figure 6—figure supplement 1: as in point 2, genetic inhibition and reporters would be very informative.

We have addressed this concern above in point 1; we have included some new data as above. However, we do not have the mutants available to perform further analysis.

4) Figure 7: these results show that collagen fiber disruption impairs regeneration. However, the link between these results and the main finding of the study, i.e. the role of vimentin in wound healing, is not clear. It would be valuable to use the vimentin reporter here.

We include a panel to show that *vimentin* reporter activity also increases at a burn however the morphology of the cells is dramatically different (Figure 7). Future work will be focused on characterizing the behavior of these cells in the context of a burn as compared to tail transection.